# Is It All about a Science-Informed Decision? A Quantitative Approach to Three Dimensions of Justice and Their Relation in the Nuclear Waste Repository Siting Process in Germany

**Lucas Schwarz** 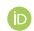

Research Center for Sustainability, Otto-Suhr Institute for Political Science, Freie Universität Berlin, 14195 Berlin, Germany; lucas.schwarz@fu-berlin.de

**Abstract:** Nuclear waste management is a contested challenge that lasts for decades. Especially in Germany, the history of the usage of nuclear energy is conflictive and notions of justice are therefore omnipresent in the ongoing site selection process for a nuclear waste repository. Against the background of injustices caused by the deployment of nuclear energy, such as the obligation for current generations to deal with nuclear waste, questions of how to justly deal with nuclear waste and to find a just repository site arise. By conducting a survey among people that participate in the site selection process as well as people living in or representing an area that is still considered suitable, the assessment of different aspects of justice was evaluated. The role of a science-informed site decision without any political bias is considered highly important for a just site selection. Distributional aspects, such as notions of utilitarianism, retribution, or the exemption of environmentally burdened regions are generally not approved but more detailed questions have shown that such notions cannot be dismissed at this early stage of the site selection process. The difference for general agreement can also be observed for intergenerational recognition, as the recognition of future generations is regarded as necessary, but concrete implications (retrievability or enclosure) are assessed ambiguously. Although some factors of justice are assessed more importantly than others, the analysis has shown that the interrelations between the different dimensions of justice are manifold and the argument that one dimension can be substituted for another one is too reductive.

**Keywords:** justice; quantitative survey; nuclear waste; repository; perception; science-informed

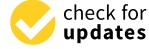

## 1. Introduction

Even more than six decades after the commissioning of the first nuclear power plant [1], nuclear energy still faces the challenge of nuclear waste disposal [2]. Worldwide, there is no operational repository for highly radioactive waste (high-level waste; HLW) that is capable of safely encapsulating spent nuclear fuel. Such products continue to radiate between 10,000 to 1 million years [3], thus adding a humanely unimaginable time scale to the complexity of depositing nuclear waste. Finland is currently constructing an HLW repository with a planned deposition date in 2025 [4]. Meanwhile, repository sites have been proposed in Sweden and Switzerland. In Germany, the 'Federal Company for Radioactive Waste Disposal' (BGE) announced that 54% of the country are potential areas for a repository site (sub-areas; areas that cannot geologically be excluded for being suitable for hosting a nuclear waste repository; regardless of above-ground land use) [5]. As legally determined, the German repository will need to be safe for about one million years, and a site decision shall be made in 2031[1]. Additionally, nuclear waste needs to be retrievable for 500 years after the initial sealing of the repository [6]. Due to the decision to phase-out the remaining three nuclear reactors in Germany, future generations[2] will not benefit from this energy source anymore. From today's technical viewpoint the waste will remain hazardous for centuries, even millennia.

This results in inequalities within and between societies and future generations as well as different spatial areas [7]. Genske [8] refers to a Kantian sense of justice (categorical imperative) stating that it was unjust to deploy nuclear energy as long as there was no solution to the waste problem. Despite this injustice, Ott and Semper [9] notice a shift towards the question of how the undeniably existing nuclear waste should be dealt with.

Discussions about justice remain mostly theoretical and vague, without a transfer to the actual challenge of finding a repository site that can be tolerated by society. Despite the introduction of the legal foundation of the German site selection process ('Repository Site Selection Act' (*Standortauswahlgesetz*)) [6] notions of justice remain imprecise as only little knowledge exists about how people perceive justice for dealing with HLW. An amalgamation of the existing literature and empirical research about the perception of justice regarding an HLW repository is necessary to provide valuable insights for the ongoing site selection process. This article thematizes the following question: Which aspects of justice are particularly important for the German repository site selection to be perceived as just? I assume that a feeling of justice can arise in the context of an HLW repository. To approach this question, the focus lies on three dimensions of justice, namely procedure, distribution, and recognition [10]. By employing this approach, this study is situated in the field of environmental and energy justice. As authors within both domains have qualitatively contributed a wide range of considerable aspects that are relevant to justice, this study shows how justice research about a so-called public bad [11] can be carried out quantitatively and what implications require special emphasis. Additionally, this study contributes to the existing literature by testing whether certain aspects that have been discussed theoretically (e.g., the role of retributive argumentation in justice) are perceived by survey respondents.

A quantitative survey of the public is designed as a fundamental acquisition and therefore covers a wide range of aspects related to justice. Although nuclear waste management has been chosen for the empirical background, such insights can potentially be transferred to other fields of justice research that thematize the management of public bads [11], e.g., atmospheric or environmental pollution. This is obvious as there is no such thing as justice that only applies to nuclear waste management, but dimensions of justice that are applicable to nuclear waste management.

Existing literature was analyzed to synthesize aspects of justice that are relevant to the case study of the German site selection process. Based on this review, a quantitative survey was carried out. The results of this survey are presented, followed by a discussion of the findings with existing literature. Some concluding remarks will show findings and needs for future research.

## 2. State of Research

The German case has been chosen due to its conflictive history and unique decision to phase-out nuclear energy in the aftermath of the Fukushima accident in 2011 [12]: For decades the German nuclear politics enforced its will against the will of the general population in a top-down manner (decide-announce-defend approach) [13,14]. Especially the designation of Gorleben as a site for a nuclear waste repository, sparked protests that were fought out in courts. While citizens employed bottom-up approaches and presented scientific studies, the state employed scientists (e.g., geologists) to support its agenda [15]. The decision to phase-out nuclear energy set a new frame for nuclear waste governance in Germany, the site selection process for an HLW repository was initiated to be comparative. Throughout German nuclear history, matters of justice were constantly fought out and therefore continue to play a major role in the newly initiated site selection process. The process is now mainly carried out by the operator BGE and the regulator BASE (*Bundesamt für die Sicherheit der nuklearen Entsorgung*, Federal Office for the Safety of Nuclear Waste Management). Citizens can get involved voluntarily by participating to discuss methodological or participatory questions.

One remark is necessary: The German site selection process is aiming to find 'a [single] site for a nuclear waste repository with the best possible safety' [6]. Although this legal

statement seems rather just, the relation between 'best possible safety' and 'most just' is rather ambivalent. There is no coherence between the two aspects, as the site with the 'best possible safety' does not necessarily need to be just. The other way around the 'most just' site does not need to be the site with the 'best possible safety'. Justice does therefore not equal safety, this distinction needs to be kept in mind for the rest of the article.

The following section thematizes the aforementioned dimensions of justice (procedure, distribution, recognition) and shows their meaning and transferability to the German site selection process. There is a large body of literature dealing with justice in general, nuclear waste management is only a niche within this domain. For this review, studies analyzing nuclear waste management, as well as studies from adjacent fields have been considered. The three dimensions of justice have been chosen following the environmental justice perspective, as presented by Schlosberg [10]. Environmental justice scholars argue that it is not sufficient to only consider one dimension of justice and therefore apply the triad of procedure, distribution, and recognition by showing their constant interrelation. As an empirical approach was chosen for the study, the literature review contains studies that do not necessarily emphasize abstract concepts of justice but rather perspectives of perception of justice.

*2.1. Procedural Justice*

Thibaut and Walker [16] present a much-cited study that shows the interplay between procedures and the justice perception of their outcomes. They have empirically shown, that a just procedure can positively increase the justice perception of a matter. Generally, procedural justice is regarded as a corrective for technocratic or uneven decision-making processes [17].

Transparency and participation are vital aspects of procedural justice. From a definitory point of view, transparency comprises the comprehensible communication of actions, thoughts, documents, data, and results [18], in ongoing procedures, however, the theoretical and practical claims of those involved can diverge. Participation is similarly ambiguous: Whereas the general idea of involving different perspectives from different parts of society seems beneficial, implementation is often lacking the sensitivity for power relations, legitimizing tactics, or matters of exclusion. Wilk and Sahler [19] describe for German civil movements how participatory procedures were used as a mean to legitimize outcomes by including civic actors. Sundqvist and Elam [20] (p. 198) describe this as *"public involvement designed to circumvent public concern"*. In the sense of Lukes [21] this can be understood as hidden power. Elam and Sundqvist [22] state for the Swedish case that the flexibility of the authorities was one of the main reasons for the success of their nuclear waste governance. This contradicts the necessity to exert (hidden) power.

Additionally, different means of power are often employed by involved actors in participatory processes. A sensitive approach is necessary to evaluate how such power relations influence the perception of justice. Arnstein [23] introduced the 'ladder of participation' as a first approach to categorize the actual power that citizens have in participatory process, but it does not aim to understand how power relations can contribute to a more just perception of outcomes. Partzsch [24] provides a triad of power relations, namely 'power over', 'power to', and 'power with'. She argues that 'power with' proposes an adequate and equal power relation between state actors and (civil) society. Schwarz et al. [25] have applied this concept to the German site selection process and conclude that a relation between stakeholders on equal footing with equal possibilities enables a procedure with an outcome that can be tolerated. Nonetheless, their hypothesis remains untested and therefore requires empirical insights on if and how power asymmetries influence the perception of justice. Chilvers and Burgess [26] provide a critical perspective on the relation between science and participation: They argue that deliberative modes of participation of scientific question bear the potential to undermine the credibility of such processes, whereas the opposite is often claimed. Additionally, in the sense of procedural justice, they call for a specific role of participation in such processes, as a corrective, that exposes vested interests.

The importance of procedural justice is stressed by Krütli et al. [27]: Based on a vignette analysis the authors state that matters of procedural justice for placing contested infrastructure are more important than matters of distributive justice. Sundqvist and Elam [20] emphasize that participation often focusses to strongly on procedural matters instead of using participation to articulate matters of legitimacy or concern. If matters of procedural justice shall contribute to a just perception of nuclear waste management, such issues need to be addressed.

To achieve this equal relationship between stakeholders, the consideration of matters of communication is necessary. Habermas [28] defines comprehensibility, truth, appropriateness, and sincerity as validity conditions for deliberation. Chang and Zhang [29] draw on this tetrad and stress the importance of honesty for the 'rightfulness of deliberative policymaking' and procedural justice. Their study provides interesting evidence that might be reproducible in the German case, which requires testing. A working definition of procedural justice for the following analysis is that procedural justice defines a procedure to be transparent, comprehensible, open for participation, and honest, while maintaining balanced power relations.

### 2.2. Distributive Justice

Distribution and procedure are strongly intertwined when deliberating question such as *"who gets what, when, and how?"* [30] (p. 20). Distribution comprises questions of spatiality as well as distribution of goods, burdens, and compensations. In Germany the equal distribution of HLW in a repository is prohibited by law, due to the aim to only construct one single repository. Claims of just distribution still require consideration [31]. As nuclear waste is associated with danger to health, albeit the low probability of an accident within a repository, the feeling of affectedness needs consideration for assessing the perception of justice [32].

Although a spatially equal distribution of justice seems impossible, different concepts of distributive justice can be applied: Utilitarianism suggests that a repository would be located in a geologically suitable region that affects the least amount of people, thus benefiting the largest share of the population [33,34]. Although classically applied to human actors, this concept can be applied to environmental aspects as well, raising the question if all environmental burdens (including a repository) should be shifted to one region to benefit all other regions. From a historical perspective, decision-makers employed the utilitarian principle by placing unwanted nuclear facilities close to rural, national borders, e.g., in Gorleben. Such remote places were thought to be simple to implement, as the regulating authorities did not expect resistance [35,36].

The concept of Retributive Justice [37] is closely related to the 'polluters pay principle' [38] and describes distribution based on who caused a burden, i.e., nuclear waste. Retributive justice states that an entity that caused a situation, should also be responsible for solving this situation. Retribution comprises the notion that nuclear communities, such as the sites of (in)active nuclear power plants, that profited over decades via taxes, revenues, and jobs, shall also host a repository, if the geological formations are suitable. The counter position could argue that such communities already served their share of responsibility and should therefore be excluded from the site selection process. This argumentation is often voiced by politicians, especially between the Eastern federal states and the Western federal states. The North Saxony District Office [39] argues that structural change and conformation between the federal states cannot result in a repository site in the Eastern federal states. Lersow [40] argues that only 3.6% of the share of CASTOR casks that contain HLW from nuclear power plants were generated in nuclear power plants from the Eastern federal states. Although such argumentations are rather general they are already observable in the German site selection process.

Another aspect of distribution that is not necessarily spatial but equally controversial is Compensatory Justice. It comprises the downstream distribution of remedies and the minimization of injustice by providing (i)material resources [41]. The compensatory

approach is generally criticized for being used as a response to bribe unwilling people, although this neglects structural dependencies and potential redemption. Sjöberg and Drottz-Sjöberg [42] showed for the Swedish repository how this was the case for a minor share of the population within the repository community. Kunreuther et al. [43] (p. 469) state that *"the public needs to be convinced before compensation is considered"* by drawing on insights from US nuclear waste management, thus showing the connection between procedure and distribution. The examination of the German case can therefore help consolidate knowledge about the perception of distributive justice. Due to its ambivalent character and different approaches compensatory justice requires empirical insights. A working definition of distributive justice for the following analysis is rather imprecise, as the individual forms of distributive justice postulate different distributions. Distributive justice defines the just distribution of infrastructure based on certain ideals with the aim of burdening as little as possible and if necessary compensating such burdens.

## 2.3. Justice as Recognition

The third dimension of justice is recognition. Honneth [44] names fundamental aspects of recognition as love, equal treatment in law, and social esteem, thus referring to recognition beyond personal or societal biases. Recognition comprises valuing individuals without any bias based on race, class [45], education, social milieu [46], affiliation, or ideas. Besley [47] and Bies [48] frame justice as recognition as interpersonal and interactional fairness, thus referring to the same basis of proper recognition to enable just procedures. Bowrey [49] (p. 11) stresses its importance as *"those without political representation can have nuclear waste forced on them against their will"*. Recognition is therefore not only about enabling all voices to be heard but to critically reflect whose voices are not heard and what keeps marginalized voices from being heard [50].

One pillar of recognition is equal treatment. Fricker [51] transferred this notion to practices of knowledge generation. She detects that recognition is two-folded, either testimonial (individuals need to be regarded as credible when contributing) or hermeneutical (individuals need access to generate knowledge). Some discussions revolve around the role of lay people in contrast to experts in such technical contexts: Aitken [52] provides an empirical example how lay knowledge was discredited in the context of wind energy, Bell [53] provides a similar study for the nuclear waste context, whereas Wynne [54] shows how lay and expert knowledge can be equally worthy and contributory to a procedure and its outcome. Bell [55] (p. 165) adds to this discussion by stressing the importance of the mobilization of 'local knowledge for more localized practices'. The question remains what kind of knowledge is recognized and what kind of actors can epistemically contribute. Geological questions are often characterized by scientific dissent [56] and uncertainties [57,58], however the societal handling of such and how it influences the perception of justice has not yet been sufficiently investigated. Marginalized people are usually overlooked in discussions about nuclear waste management, as the studies by Hurlbert and Rayner [59] or Nowlin and Conner [60] have shown. However, this exclusion can also contribute to the fact that important, local knowledge is not put to use, which (again) raises the question of just treatment.

Due to the radioactivity and the long-time spans of nuclear waste management, intergenerational matters require consideration [61,62]. Schlosberg [10] argues that future generations have to be able to act within the same margins of freedom as current generations. Kermisch [63] assesses in her study, what kind of repository can be the best for future generations and concludes that a non-retrievable deep-geological repository is the best solution in the long term (safety-wise), as future generations do not necessarily have to deal with the waste anymore. Interests of future generations remain hypothetical though, as direct reciprocity in current procedures is impossible. Nonetheless, intergenerational argumentations have gained interests in public discourse, especially due to climate change and sustainability debates [64]. Current generations try to assess the needs of future generations but often do so very differently. It is therefore necessary to know how the

actual implementation of intergenerational matters correlates to the perception of justice. A working definition for justice as recognition for the following analysis is that recognition requires the equal treatment of participants, regardless of any given characteristics or social position, while not only enabling each actor to speak but also to be properly heard.

None of the three dimensions of justice can solely account for the perception of justice regarding nuclear waste management. As the German search for a nuclear waste repository was characterized by strong civil opposition in the past, it is worth to include adjacent factors such as trust, perception of risk, and emotions.

### 2.4. Adjacent Factors

In the context of nuclear waste management, the influence of various factors has been investigated empirically. Trust was given a great deal of research attention. Slovic et al. [65] strongly emphasize the role of trust in relation to perceived risk, as they state that public opinion about nuclear waste facilities is influenced by 'images of fear', thus impeding a trusting environment. In more recent work, Lehtonen et al. [66] ascertain that the German society is a 'society of mistrust' and conclude that this provided the basis for the development of counter-expertise in the German repository site selection process. Choi and Matsuoka [67] study the relation of trust, distributive, and procedural justice and conclude that there is a changing and interactive relation between the three based on the national circumstances (e.g., phase-out decision) and local contexts (e.g., proximity to repository sites for low level waste (LLW) or intermediate level waste (ILW) and nuclear power plants). Seidl et al. [68] conduct a baseline study about trust and nuclear waste management in Germany and ascertain that trust determines acceptance of a nuclear waste repository. Nonetheless, they do not elaborate on the relation to the perception of justice. As the above-mentioned studies have exemplarily shown, trust seems to play a non-neglectable role in nuclear waste management, its influence on the perception of justice in the German case requires additional empirical insights. The above-mentioned studies from different contexts are therefore taken as basics for the survey, thus it is tested if the empirical insights from Germany can add to this rich body of literature by verifying earlier results.

Another factor that has been given attention are emotions. Roeser [69] provides insights from debates about nuclear energy and risks and shows that the attribute 'emotional' is often equated to 'being irrational' and therefore used as a mean to discredit the liability of personal opinions or statements. In this context Slovic et al. [65] include dread of nuclear facilities in their analysis of risk perception. Sjöberg [70] adds that the reduction of emotions to affect is counterproductive to risk communication and therefore the consideration of emotions to accommodate concerns is necessary. Additional factors that have to date only been theoretically assessed or in different contexts, but still need empirical observation (as usual in relation to the perception of justice) are general opinions towards nuclear energy and the German phase-out decisions as well as the prerequisites of the site selection process [12], personal experiences [71,72] and spatial effects [73].

Considering the various studies in this section, it is apparent that insightful work around the perception of justice for nuclear waste management has been carried out. Nonetheless, most work has been conducted in a theoretical or qualitative way. It is therefore necessary to conduct a study that covers fundamental assessments to test how results from other case studies are transferable and how the empirical insights compare or add to existing contributions.

## 3. Materials and Methods

In order to answer the research question 'Which aspects of justice are particularly important for the German repository site selection to be perceived as just?' a quantitative, Germany-wide survey was conducted. The survey comprised the three dimensions of justice (procedure, distribution, recognition) by utilizing operationalized statements.

### 3.1. Operationalization

As shown in Section 2 some insightful work has already been carried out. Nonetheless, some research potential remains regarding the transferability of results from case studies from different national contexts and regarding the comparability of theoretical contributions and empirical insights. The three dimensions of justice were operationalized by splitting them into aspects (items) as indicated by the previously mentioned studies. To simplify the level of abstraction, operationalized statements were formulated for each identified aspect of justice. The survey participants were then asked to assess each item on a scale from 0 (lowest) to 5 (neutral) to 10 (highest), depending on how much they agreed with a statement. In addition to the statements about perceptions of justice, participants were asked to give some general insights on their stance towards nuclear energy, the nuclear phase-out decision of the German government [12], as well as whether they have lived (or grew up) in the vicinity of a nuclear power plant or interim storage site. The operationalization of the applied statements as well as their derivation from previous theories or studies is presented in Table 1. For the formulation of the statements, negative formulations were avoided to focus on justice instead of injustice. Finally, some comparison questions were included in the questionnaire to compare the different aspects of justice.

**Table 1.** Conceptual Framework for the Quantitative Survey.

| ID * | Dim. | Factor | Definition/Background | Conceptualization (Translated) | Reference |
|------|------|--------|----------------------|-------------------------------|-----------|
| PJ1 | Procedural Justice | Transparency | Constant disclosure of procedural actions and documents, based on accountability, openness, and efficiency | A just site selection process is transparent. | [18,47] |
| PJ2 | | Comprehensibility | Use of understandable language | A just site selection process is comprehensible. | [28,29] |
| PJ3 | | Participation | Possibility to bring arguments meaningfully into a process, independent of one's position | A just site selection process enables participation. | [23] |
| PJ4 | | Honesty | Truthfulness by speaking actors that is regarded as trustworthy be receiving actor | Mistakes should be openly communicated. | [28] |
| PJ5 | | Power | Possibility for one actor to overrule another actor | All actors should have the same opportunities to influence. | [24] |
| DJ1 | Distributive Justice | Affectedness | NIMBY (Not in My Backyard)—People tend to agree to infrastructural changes as long as those are not in their vicinity | If my region is geologically the most suitable for a repository, I agree to the construction of a repository there. | [32] |
| DJ2 | | Utilitarianism | Greatest possible benefit for the greatest amount of people | The final repository site is more just if fewer people are affected by it. | [33,34] |
| DJ3 | | Compensation | Providing (i)material resources to minimize inequality, such as risk or loss | The repository community is entitled to generous financial compensation. | [41–43] |
| DJ4 | | Retribution | Response to harm, 'polluters-pay-principle' | If the geology is suitable, a region that has benefited strongly from nuclear energy should also host the repository. | [37,38] |
| DJ5 | | Environmental Burden | Exemption of environmentally burdened areas from the site selection process (utilitaristic environment-centered perspective), e.g., sites with heavy industry or environmental pollution | The search for a repository should not include regions with a high environmental burden. | [34] |
| IJ1 | Justice as Recognition (Intergenerational) | Future Generations | (Hypothetical/Anticipated) Inclusion of Wants and Needs of Future Generations | Future Generations have to be considered in the siting procedure for a repository. | [61,62] |
| IJ2 | | Young Generations | Young generations as a constant mediator to future generations | Intergenerational justice comprises the inclusion of the younger generation. | [25] |
| IJ3 | | Reversibility | Retrievability of nuclear waste | For the sake of future generations, the repository should be kept open | [63] |
| IJ4 | | Closure | No points of contact with nuclear waste to be able to respond to future challenges | For the sake of future generations, the repository should be sealed. | [63] |
| IJ5 | | Timely Solution | The legal definition of how long the procedure to find a repository site will last, predictability | A final repository must be found quickly so that future generations are not burdened. | [6] |

| ID * | Dim. | Factor | Definition/Background | Conceptualization (Translated) | Reference |
|---|---|---|---|---|---|
| RJ1 | | Equal Treatment | Equal treatment of all actors (without any societal, testimonial, or personal bias) | All participants must be treated equally in the site selection process. | [44–46] |
| RJ2 | | Expert Knowledge | Importance of expert knowledge | Expert knowledge must be recognized in the site selection process. | [52,54] |
| RJ3 | | Lay Knowledge | Importance of laypeople's knowledge | Lay knowledge must be recognized in the site selection process. | [52,54,56] |
| RJ4 | | Uncertainties | Communication of geological, societal, and other scientific uncertainties | Communicating scientific uncertainties is important to me. | [57] |
| RJ5 | Justice as Recognition (Current, Epistemic) | Dissent | Communication of geological, societal, and other scientific dissents | Communicating scientific dissent is important to me. | [56] |
| RJ6 | | Access to Independent Studies | The necessity to enable access to independent studies by external scientists (hermeneutical capability) | Access to independent studies is important to me. | [51] |
| RJ7 | | Independent Control Institution | The necessity to involve independent control mechanisms, such as peer-review procedures by external scientists | Independent, external process control is necessary. | [26] |
| RJ8 | | Process Length | Reasons to expand the procedure's length | The process may take longer than planned. <br><br>(a)  Scientific findings take time. <br>(b)  Participation takes time. | [6] |
| AF1 | | Trust | . . . in the scientific foundation | I trust that science will find the best possible final repository site. | [67,68] |
| AF2 | | | . . . in the technological barrier | I trust that technology will enable the best possible repository site. | |
| AF3 | Adjacent Factors | | . . . in the geological formation | I trust that geology will find the best possible solution for a repository. | |
| AF4 | | Emotions | Importance to include fears and other emotions in deliberation | Scientific arguments are more important than fears expressed. | [69] |
| AF5 | | | | Scientific arguments are more important than emotions expressed. | |
| GS1 | | Risk | Personal assessment of risk for a nuclear waste repository | A repository is a high-risk facility. | [69] |
| GS2 | | Phase-Out | Personal assessment of whether the phase-out decision was right | Phasing-out nuclear energy in Germany was the right decision. | [12] |
| GS3 | General Statements | Fear | Personal assessment of fear | A repository in my vicinity scares me. | [32] |
| GS4 | | German Repository | The legal foundation defines that the repository must be situated in Germany. | The storage of highly radioactive waste from German nuclear power plants must take place in Germany. | [6] |
| GS5 | | One Repository Aim | The legal foundation defines that there must be one (single) repository for all HAW from Germany. | The storage of highly radioactive waste from German nuclear power plants must take place in one final repository. | [6] |
| CQ1 | | Spatially balanced site decision | Distributive Justice | A spatially well-balanced site decision is important for a just repository site. | [74] |
| CQ2 | | Compensation | Distributive Justice | Adequate compensation is important for a justrepository site. | [32] |
| CQ3 | | Just process | Procedural justice | A just process is important for ajust repository site. | [16,67] |
| CQ4 | | Future Generations | Intergenerational Justice | The consideration of future generations is important for a just repository site. | [61] |
| CQ5 | | Flat Power Hierarchies | Justice as Recognition | A balanced influence of all actors is important for a just repository site. | [24] |
| CQ6 | Comparison questions | Scientific Site Decision | Epistemic Justice | A science-informed decision is important for a just repository site. | [51] |
| CQ7 | | Political Consideration | Justice as Recognition | Political consideration is important for a just repository site. | [35] |
| CQ8 | | Emotions | Adjacent Factor (Emotions) | The consideration of emotions is important for a just repository site. | [69] |
| CQ9 | | Time Specification | Adjacent Factor (Time) | A clear timeframe is important for a just repository site. | [6] |
| CQ10 | | Trust | Adjacent Factor (Trust) | Trust in the operator/Regulator is important for a just repository site | [68] |

**Table 1.** *Cont.*

| ID * | Dim. | Factor | Definition/Background | Conceptualization (Translated) | Reference |
|---|---|---|---|---|---|
| SA | Spatial Assessment | Land Use, Place Identity, Risk Perception | Assessment of different kinds of land use for hosting a nuclear waste repository | Provided the geology for a repository is equally suitable everywhere: Please indicate how just you perceive the following (schematic) sites.<br><br><br><br>This map exemplarily shows the schematic style of the assessed maps. Land uses: Major city, rural village, lakeside, seaside, riverside, forest, mountains, agricultural field, nuclear power plant, interim storage site, border, urban border, rural border | [71–73] |

* In the results chapter, the ID will indicate which factor was used.

### 3.2. Sample

The question of how the statistical population is constituted is challenging: Germany has ~84 million inhabitants at the moment. A total of 54% of the country's area are designated as sub-areas. Nonetheless, it is not a question for the inhabitants of those 54% only, as the repository shall host all the HLW from Germany (before 1990: East and West Germany). As Taebi [57] has vividly compared, the time scales of a repository potentially transcend the time scales of national states. She shows how the Slovenian capital Ljubljana has lain in seven different countries over the last century, thus raising the question if a repository can be regarded as a national matter only. The survey population could therefore also be multinational, if not even global.

Despite its long-term importance, nuclear waste management can be characterized as a niche topic in public discourse. In the German site selection process on average 1430 people registered for the official participatory events (sub-areas conference; Fachkonferenz Teilgebiete) that took place between February and August 2021 (online as well as in a hybrid format) [25]. This accounts for as little as 0.0017% of the German population. Representativity was not given at any moment, neither statistically nor societally, politically, or democratically. The question of who constitutes the statistical population therefore remains unclear. As the target population needs to be clear to conduct a quantitative survey, it was defined as people that are thematically involved in the site selection process, either by choice (participation), by employment (e.g., operator (BGE), politicians, NGOs), or by spatial vicinity (living in one or more of the sub-areas). The latter refinement of the German population was chosen as nuclear waste management is a niche topic and due to the simultaneity of global (e.g., environmental pollution or climate change) and national challenges, it was presumed that people how do not live in a sub-area also tend not to participate in the site selection process. This assessment is derived from data from the operator BASE: This data shows that there were proportionally more participants at the sub-areas conference from federal states (e.g., Lower Saxony and Bavaria) that contain a large share of sub-areas[3]. Albeit this focus, 36.7% of the respondents do not currently live within a sub-area (see Figure 1). Table 2 shows the composition of the sample in detail.

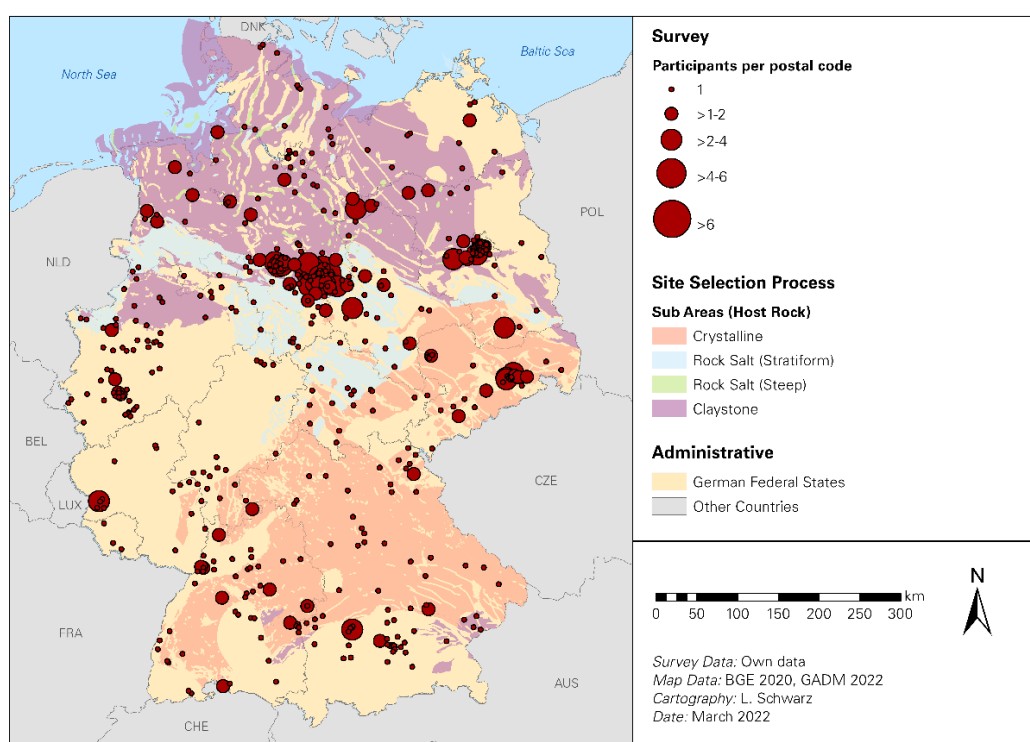

**Figure 1.** Spatial Distribution of the Survey Respondents based on the Postal Code (own data).

**Table 2.** Sample composition * (own data).

| | |
|---|---|
| Age structure | <18—0.3%; 18–29—14.4%; 30–44—28.9%; 45–59—29.9%; 60≥—19.3% |
| Process role | Citizen—44.3%; Employees of operator (BGE)—14.2%; Representatives of local authorities—12.6%; Scientists—10.1%; NGO member—5.9%; Employees of regulator (BASE)—1.0%; Others—11.9% |
| Distance to nuclear power plant | <20 km—23.0%; >20 km—71.6%; Unsure—4.1% |
| Situated within sub-area | Yes—48.3%; No—36.7%; Unsure—10.8%; Unaware—2.8% |
| Region within Germany | Berlin—6.3%; North—29.9%; East—17.7%; South—19.5%; West—14.5% |
| Participation frequency in the site selection process | Never—41.1%; Annually—42.3%; Monthly—11.2%; Weakly—4.1% |

* If the categorial sum is below 100%, the remaining share is accounted for by abstentions.

To generate insights the following approach was chosen: As nuclear waste management is a niche topic, participants from the ongoing site selection process were contacted to take part in the survey. Those participants were recruited via a mailing lists that was institutionalized after the first sub-areas conference as a networking platform for participants of the participatory event. It is managed by the regulator but not moderated. Additionally, participants were recruited based on their affectedness by the site selection process. Municipalities that overlap with a sub-area, thus municipalities that cannot be excluded for being geologically unsuitable for a repository from the ongoing process, were contacted via their local administration (contact persons in districts). Non-governmental organizations were contacted via mail and additionally the link to the survey was posted in thematically fitting Facebook groups (e.g., groups that revolve around nuclear waste, have a thematic connection (pro- or anti-nuclear energy), or deal with environmental protection). On the

national and federal level party-related working groups on energy, infrastructure, or waste management were invited to participate. A total of *n* = 716 participants were recruited over a time span of two months (August–September 2021).

### 3.3. Statistical Analysis

For evaluating the collected data, different statistical methods were applied: Next to the descriptive possibilities (e.g., mean to calculate average values), correlations and regressions were used to analyze the relation between different aspects of justice. Since the data were ordinally scaled, the correlations were calculated using the Spearman coefficient (two-sided, ρ). Relations were interpreted on the $\alpha = 0.05$ (*) and $\alpha = 0.01$ (**) level, to ensure that no arbitrary conclusions were drawn. The strength of the correlations is classified the following way: 0–0.19 = very weak correlation; 0.2–0.39 = weak; 0.4–0.59 = moderate; 0.6–0.79 = strong; 0.8–1 = very strong (cf. [75]).

The regressions were calculated using ordinal logistic regressions [76]. To determine the statistical fit of the regressions, the Goodness-of-Fit test was used [77], to determine the statistical robustness of the regression by assessing the explanatory value of the independent variables in the dependent variable. The pseudo $r^2$ value (Nagelkerke) serves as an estimate of the proportion to which the dependent variable is explained by the independent variables. Its general usability is contested though [78]. Therefore, the Nagelkerke score was used as a rough value for orientation and not as an exact measurement.

## 4. Results

In the following section, the results for each dimensions of justice will be presented, as well as their interrelation with one another and adjacent factors. In general, there is a slight tendency among the participants of the survey (the sample) that the German nuclear phase-out was the right decision (GS2; Ø = 6.74, SD = 3.87). The sample does not entirely agree with the legal foundations of the process: 'only' 50.9% strongly agree that a repository needs to be found in Germany (GS4; Ø = 7.72, SD = 3.15). Another aim of the siting procedure, the one repository aim, is assessed ambivalently: There is no clear tendency within the sample (GS5; Ø = 4.95, SD = 3.46). The ambivalent assessment of the prerequisites of the siting procedure potentially pose a difficulty regarding the perception of justice of the repository site.

### 4.1. Procedural Justice

Based on the contributions of Habermas [28], as well as more recent studies from Chang and Zhang [29], Ball [18], or Partzsch [24], the factors transparency (PJ1), comprehensibility (PJ2), participation (PJ3), honesty (PJ4), and power (PJ5) have been analyzed. Whereas transparency (Ø = 6.44, SD = 3.26), comprehensibility (Ø = 6.41, SD = 3.32), participation (Ø = 6.45, SD = 3.13), and power (Ø = 6.61, SD = 3.18) are similarly assessed, honesty is assessed as a highly important factor for the perception of justice (Ø = 9.11, SD = 1.70). Table 3 presents the correlations between the five selected factors for procedural justice. Especially, the correlation between comprehensibility and transparency is positive and very strong (ρ = 0.911 **). Additional strong correlations exist between participation and transparency (ρ = 0.755 **) as well as participation and comprehensibility (ρ = 0.755 **). Honesty does not correlate as strongly with the other factors, as it is associated with an outstanding significance. The notion of power only weakly correlates with honesty (ρ = 0.246 **), whereas there is no correlation with the other factors of procedural justice.

**Table 3.** Correlations between factors of procedural justice (own data).

| | | Transparency (PJ1) | Comprehensibility (PJ2) | Participation (PJ3) | Honesty (PJ4) |
|---|---|---|---|---|---|
| Comprehensibility (PJ2) | $\rho$ | 0.911 ** | - | 0.760 ** | 0.198 ** |
| | Sig. (2-sided) | <0.001 | - | <0.001 | <0.001 |
| | N | 687 | - | 672 | 682 |
| Participation (PJ3) | $\rho$ | 0.755 ** | 0.760 ** | - | 0.246 ** |
| | Sig. (2-sided) | <0.001 | <0.001 | - | <0.001 |
| | N | 672 | 672 | - | 666 |
| Honesty (PJ4) | $\rho$ | 0.198 ** | 0.198 ** | 0.246 ** | - |
| | Sig. (2-sided) | <0.001 | <0.001 | <0.001 | - |
| | N | 681 | 682 | 666 | - |
| Power (PJ5) | $\rho$ | 0.003 | −0.007 | −0.007 | 0.246 ** |
| | Sig. (2-sided) | 0.935 | 0.864 | 0.848 | <0.001 |
| | N | 684 | 685 | 669 | 684 |

** = Level of significance is at $\alpha = 0.01$

Honesty weakly correlates with the communication of uncertainties (RJ4; $\rho = 0.400$ **), the access to independent studies (RJ6; $\rho = 0.401$ **), and the necessity for an independent process control (RJ7; $\rho = 0.410$ **). Honesty can be understood as a non-neglectable aspect of procedural justice, that is embedded within the science-informed approach to finding a suitable site for a repository.

*4.2. Distributive Justice*

For the justice dimension of distribution, participants assessed the factors affectedness (DJ1), utilitarianism (DJ2), compensation (DJ3), retribution (DJ4), and environmental burden (DJ5). As NIMBY attitudes are potentially observable in the case of nuclear waste management [14], participants assessed whether they would approve of a repository in their vicinity, given the geological suitability. A total of 44% strongly approve of a repository under such circumstances (DJ1; $\emptyset = 7.87$, SD = 2.80). Only 5% of the participants strongly oppose a repository under such circumstances. This implies a certain capacity of abstraction, as the repository is a deep-geological facility, in contrast to everyday, above-ground life. It is especially interesting, that there is a weak negative correlation between the site approval and utilitarian thoughts ($\rho = -0.330$ **): A stronger approval by the participants correlates to a decrease in the importance of land use (regarding whether a repository would be in an urban or rural area, thus influencing the number of affected people). As long as the geological formation is arguably the best possible formation, above-ground land use is not considered as important anymore. The same finding is observable for the correlation between site approval and environmental burden ($\rho = -0.339$ **). This approval can be explained by general trust in geological formations (DJ1*AF3; $\rho = 0.407$ **), as well as trust in scientific information (DJ1*AF1; $\rho = 0.360$ **), and technologies (DJ1*AF2; $\rho = 0.316$ **). The approval negatively correlates with the assessment that a repository is a risky facility (DJ1*GS1; $\rho = -0.303$ **).

Utilitarianism argues that a repository site is just if the greatest amount of people benefits, thus burdening only a small share of a population. This does not generate great support among the participants (DJ2; $\emptyset = 3.91$, SD = 3.45) and weakly correlates with fear of a repository site in one's vicinity ($\rho = 0.340$ **). Additionally, utilitarianism is not limited to the affection of people but also to environmentally burdened places (DJ5). A moderate correlation between utilitaristic (people-oriented) argumentations and the exemption of environmentally burdened places is apparent ($\rho = 0.425$ **).

Different factors of distributive justice gained different approval from the participants: Utilitarianism, exemption of environmentally burdened regions ($\emptyset = 3.42$, SD = 3.30), and retribution (DJ4; $\emptyset = 5.58$, SD = 3.36) rank lower than the approval of compensation (DJ3; $\emptyset = 6.99$, SD = 3.00) as a necessity for a just site selection. Retribution is especially contested, as 14.22% strongly disagree that this is just, while 17.53% strongly agree that this enhances justice for the site selection. While retribution would seem a valid notion from a distributive

perspective, it neglects notions of proper recognition, as certain communities would be reduced to their nuclear installations without considering responsibility, exclusion or risk. Taking environmental burdens into account is generally not regarded as necessary for a just site selection (Ø = 3.42, SD = 3.30).

For the spatial assessment (SA, Figure 2) sites that fall within retributory argumentations, such as (in)active nuclear power plants or interim storage sites are perceived as most just (Ø = 7.67/7.74). In contrast to this, the sea (Ø = 3.78, SD = 3.46), rivers (Ø = 4.20, SD = 3.41), urban borderlands (Ø = 4.81, SD = 3.29), and urban areas in general (Ø = 4.86, SD = 3.35) are perceived as the least just sites. Interestingly though, no site has been fully disapproved, as the lowest average score ranks at Ø = 3.78 and no site has been fully approved, as the highest average score ranks at Ø = 7.74.

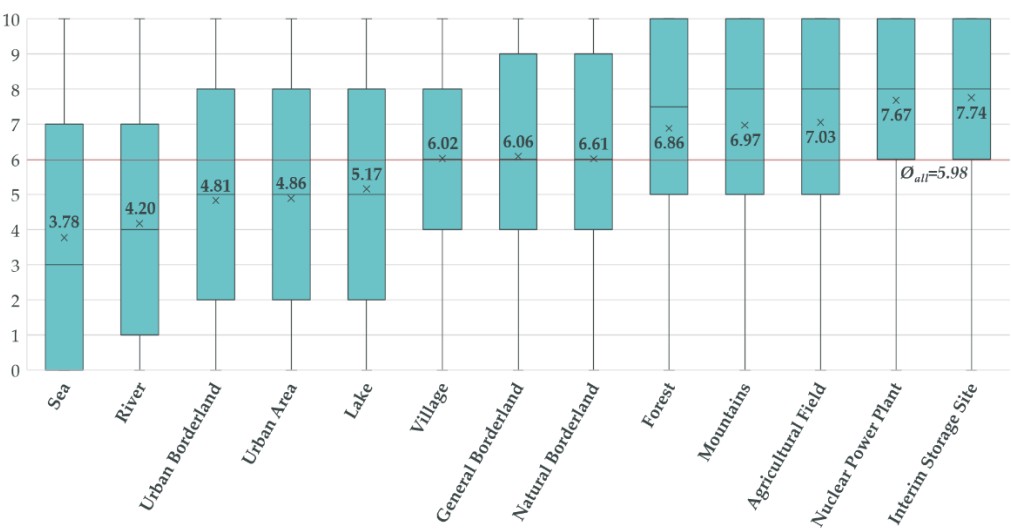

**Figure 2.** Assessment of different spatial site scenarios (SA; own data).

### 4.3. Justice as Recognition

The last dimension of justice within Schlosberg's [10] environmental justice framework is recognition. To properly approach recognition, the following results are presented in the sections of intergenerational justice, interpersonal justice, and recognition of knowledge.

### 4.3.1. Intergenerational Justice

For the recognition of future generations, the participants consent that future generations have to be considered (IJ1; Ø = 8.31, SD = 2.71). This need for recognition correlates with the necessity that young generations nowadays need to be represented in processes (IJ2; ρ = 0.548 **), the risk assessment of a future repository (GS1; ρ = 0.362 **), as well as the statement that the phase-out of nuclear energy was the right decision (GS2; ρ = 0.340 **). When assessing more precise statements, the tendencies shift: Two statements were formulated about whether to keep the repository open to grant future generations flexibility (IJ3; Ø = 5.14, SD = 3.39) or to close it so that future generations do not have to deal with the repository anymore (IJ4; Ø = 5.06, SD = 3.42). The assessment of those statements is ambivalent. For both options (open/closed repository) the neutral option is the most chosen one (25.12%/25.43%) and the extremes strongly disagree (15.04%/17.44%) and strongly agree (16.50%/16.86%) are comparatively frequently chosen. There is no clear tendency to see which of the two options is preferred by the participants to justly deal with HLW from an intergenerational perspective. Retrievability and closure negatively correlate very strongly, so there is almost no personal indecisiveness apparent in this aspect (ρ = −0.832 **). Closure additionally correlates with the necessity for a timely solution for a repository site (ρ = 0.328 **). The statement about a timely solution, as the repository might be a future burden among potentially unknown challenges, does not seem to play a major role in the assessment of justice (IJ5; Ø = 6.45, SD = 2.99).

### 4.3.2. Interpersonal Justice

To assess notions of interpersonal recognition, statements were formulated that can also be closely linked to deliberations in participatory procedures. An overlap to procedural justice is apparent. Equal treatment of all participants when dealing with HLW is emphasized by the survey participants (RJ1; Ø = 7.44, SD = 3.05). This notion correlates with the inclusion of future generations (IJ1; ρ = 0.302 **; especially the young generation (IJ2; ρ = 0.425 **)).

### 4.3.3. Recognition of Knowledge

As presented in the literature review, recognition comprises knowledge generation, e.g., by considering the relation between expert and lay knowledge. The participants assessed the importance of expert knowledge in the siting procedure regarding the perception of justice as very high (RJ2; Ø = 9.35, SD = 1.29), compared to the importance of lay knowledge (RJ3; Ø = 5.41, SD = 3.50). Additionally, trust in finding the best possible repository via scientific information is important to the perception of justice (AF1; Ø = 7.68, SD = 2.58). In this context, participants tend to emphasize rationality as there is a correlation between site approval and the importance of factual arguments compared to fears (DJ1*AF4; ρ = 0.383 **) and compared to emotions (DJ1*AF5; ρ = 0.355 **). The role of a science-informed decision for the repository site is regarded as the most important factor (CQ6; Ø = 9.35, SD = 1.18), a deeper analysis of this factor is necessary: A just procedure for a scientific solution comprises the necessity to communicate scientific disagreements (RJ5; Ø = 8.96, SD = 1.46), communication of scientific uncertainties (RJ4; Ø = 8.88, SD = 1.86), access to independent scientific studies (RJ6; Ø = 8.98, SD = 1.65), comprehensibility (PJ2; Ø = 9.00, SD = 1.71), as well as an independent control (RJ7; Ø = 9.03, SD = 1.81). A regression analysis was carried out to assess how the described factors revolving around epistemic recognition influence the justice assessment of a science-based decision. The factors communication of dissent (RJ5) and uncertainties (RJ4), access to independent studies (RJ6), comprehensibility (PJ2), independent process control (RJ7), trust in geology (AF3), science (AF1), and technology (AF2), as well as recognition of expert (RJ2) and lay knowledge (RJ3), were included in the regression. A robust solution was found (pseudo $r^2$ = 0.447), thus showing that this model has a relatively high explanatory value for a just science-informed decision. Especially the access to independent studies and trust in geology influenced the regression. If only trust in science, technology, and geological formation is considered, the explanatory value is lower (pseudo $r^2$ = 0.278).

### 4.4. Comparison Questions

Finally, the participants had to assess the most important aspects of a just site selection for a nuclear waste repository (cf. Figure 3; CQ). The almost unanimously most important aspect that contributes to a just site selection is a science-informed site decision (CQ6; Ø = 9.35, SD = 1.18) followed by a just procedure (CQ3; Ø = 8.57, SD = 2.20). The least important factors that do not contribute as strongly to a just site selection are the consideration of emotions (CQ8; Ø = 5.15, SD = 3.18) and political considerations (CQ7; Ø = 4.01, SD = 3.33). A just process (CQ3) correlates with the recognition of future generations correlate (CQ4; ρ = 0.428 **), compensations (CQ2; ρ = 0.310 **), a spatially balanced site decision (CQ1; ρ = 0.533 **), the consideration of emotions (CQ8; ρ = 0.238 **) and influence by all actors (CQ5; ρ = 0.514 **).

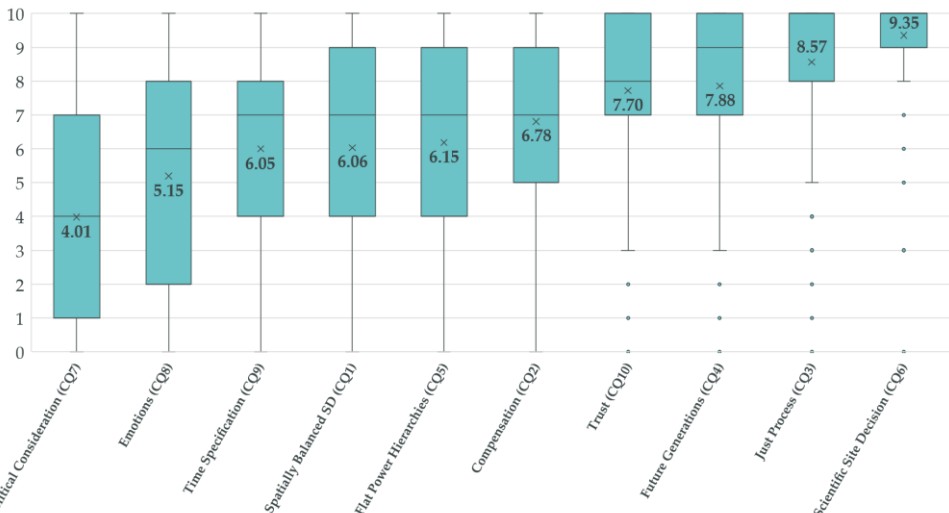

**Figure 3.** Factors contributing to a just perception of the repository site selection (CQ; own data).

## 5. Discussion

The survey of the three dimensions of justice (procedure, distribution, recognition) provides a detailed perspective on the perception of justice for HLW management in Germany. Additionally, based on the results presented in Section 4 further questions arise that require discussion. In this chapter the three following dimensions will be discussed: How can the presented results be placed within the current state of research? Can the scientificity of the site selection process be considered as the most relevant determinant of the perception of justice? What other research needs arise due to the constitution of the sample?

### 5.1. How Can the Presented Results Be Placed within the Current State of Research?

Results regarding aspects of procedural justice mostly fit to previous research, also from other national contexts. Previous work by Habermas [28], Chang and Zhang [29], as well as Ball [18], and Partzsch [24] stated that transparency, comprehensibility, participation, honest, and power are major factors that determine if deliberative processes can be successful and contribute to the quality of the outcome. As the aspect of justice had a low importance in their studies, it was analyzed in this one: The results show that especially honesty is highly important to the perception of justice regarding HLW management. For this survey honesty was conceptualized as 'open communication of mistakes' (PJ5), but those results need to be refined in future studies. Although, honesty and transparency only weakly correlate in this study, this does not allow conclusions regarding the quality of honesty for transparency. The result that honesty is valued with such outstanding significance for a just site selection potentially shows that past experiences have made participants cautious. Honesty as a value cannot be neglected in future studies but it needs additional qualification that exceed the abstract conceptualization that has been utilized for this fundamental assessment.

Choi and Matsuoka [67] argued if distributive justice cannot be achieved, procedural justice will fill its place. This argumentation is too reductive. One dimension of justice cannot be regarded as non-achievable, even if equal distribution cannot be achieved (due to the one-repository aim), just distribution can be achieved by other measures—unequal does not equate to unjust (cf. [79]). The results of the survey regarding the relation of procedural and distributive justice (e.g., correlation of CQ3*CQ1 or CQ3*CQ2) support the statement of Ravenhill [30] that distribution and procedure are constantly intertwined.

For the distributive aspects the participants of the survey have shown a high capability for abstraction as neither people-oriented nor environment-oriented utilitarian arguments strongly contributed to a strong perception of justice. Although this is a promising and

almost idealistic indication it needs to be regarded with care: The German site selection process is still in an early stage with an expected site decision in 2031; 9 years from now. Additionally, the participants of the survey were mostly acclimatized to the site selection process and are—as already indicated—in no way representative for a wider share of the German population. When actual affection will arise, as soon as geological surveys will begin in siting areas, this statement needs to be re-evaluated, alongside the NIMBY-phenomenon [32].

Nonetheless, some discrepancies arose for the assessment of distributive aspects of justice: The sample was generally critical of retributive justice (DJ4), meaning that places that financially profited from nuclear energy such as sites of power plants or interim storage sites that received tax revenues, etc. should host the repository. When assessing which site (SA) would be a just repository site given the geological suitability, there was a clear tendency towards sites of nuclear power plants and interim storage sites. This result needs additional insights regarding the motivation to assess those places as more just than the vicinity of an urban area or a rural village. One explanation can concern the transportation safety of HLW, thus effectively minimizing risks of accidents while transporting HLW to a spatially close repository site. The other explanation could relate to retributive arguments: If those communities profited from nuclear energy and they are geologically the best site, they should host the repository as well. This is highly contradictory though, as the participants assessed that retributory notions do not contribute to a strong perception of justice: Generally speaking the participants rejected retributory justice, but given a practical example (such as repository site next to a nuclear power plant or an existing interim storage site) the participants indirectly favored retributory notions. This assessment can be based in other aspects though, such as minimization of transport-related risks. For clarification additional observations are necessary as the second explanation would challenge notions of proper recognition, as habitants of nuclear communities would be reduced to their vicinity to nuclear installations—this vicinity can be arbitrary though, e.g., through birth.

Especially regarding the necessity to include the potential needs of future generations the participants of the survey showed a clear tendency towards recognition. This is congruent with the observations made by Skillington [64]: She showed for the case of climate change, that intergenerational aspects strongly matter in climate change discourses. The same can be said for HLW discourses, due to the long-time span of the challenge. Therefore, Kermisch [63] clearly concludes that a closed deep-geological repository is the best solution—and therefore the most just—for distant future generations. When assessing whether a repository should be held open for flexibility or closed so that future generations do not have to deal with it anymore, the sample is evenly divided. Although it is acknowledged that the needs of future generations have to be recognized, the actual way to implement this is highly contested. This is a potential area of conflict. The German legislation states that the repository has to be kept open for 500 years after the initial storage of HLW in the repository. Nonetheless, this law also states that the procedure is subject to the actual knowledge generated in scientific procedures and is therefore subject to change.

The most striking aspect of justice is that the participants assessed that scientificity plays a role of utmost importance for the perception of justice when dealing with HLW in Germany. Scientificity was conceptualized in multiple ways: (1) Expert knowledge must be recognized in the site selection process (RJ2); (2) The process may take longer than planned, as scientific findings take time (RJ8); and (3) A science-informed decisions is important for the best repository site (CQ 6). Especially, the last statement was unequivocally assessed (cf. Figure 3). The high importance of expert knowledge supports the findings of Aitken [52] but contradicts the findings of Wynne [54]. This clarity raises the question whether scientificity can be considered as the most relevant determinant of the perception of justice in the repository site selection process.

### 5.2. Can the Scientificity of the Site Selection Process Be Considered as the Most Relevant Determinant of the Perception of Justice?

An overly strong focus on the importance of a science-informed decision contradicts the work of Young [80] and Schlosberg [10], who argue in favor of a balanced approach to justice that is comprised by multiple dimensions that contribute to the overall assessment. This is problematic as essential parts of knowledge generation are neglected: The German site selection process is regulated by the state and operated by a state-owned federal company. The operator BGE and the regulatory authority often act as scientific actors by publishing research calls, by funding research projects, and by participating in research activities (e.g., joint laboratories). The state is therefore giving up its neutral position by actively engaging in research activities. By funding certain projects path dependencies can occur that can have a negative effect on the perception of justice in a later stage of the process. State actors will need to act towards the inclusion of other types of knowledges (such as academic knowledge from universities or local knowledge) to enable a more robust solution. A mere focus on state knowledge neglects participation as a corrective [17] and therefore potentially challenge the justice perception of the repository site.

Fricker [51] has vividly shown the potential for injustice in epistemic processes, either by testimonial (discrimination because of external characteristics, e.g., skin color) or by hermeneutical injustice (systematic denial of access to knowledge). While testimonial injustice is often based in stereotypes, misrecognition, and interpersonal interactions, hermeneutical injustice is based in structural injustices. If scientificity is of utmost importance for a just repository site, structural shortcomings need to be addressed, such as barrier-free access to institutions of knowledge generations. Every participant or citizen in the procedure needs to be enabled to understand why decisions were taken, why research projects were initiated, or what the scientific base for the final site selection will be. Otherwise citizens cannot fulfill their role as a corrective for technocratic or uneven decision-making processes [17].

The role of a science-based decision is highly important for the site selection. Without a scientific base, a decision for a repository site is not possible. Yet, geological inquiries are usually characterized by a high degree of uncertainty until the underground explorations take place. Marsily et al. [58] raised the questions whether geologists can guarantee the isolation of nuclear waste and argues that this cannot possibly be the case. Scientific information will be an essential part of a just site selection process. In the German case this was not always the case though, as the elimination of the Gorleben salt dome form the current site selection process has recently shown [81]: After decades of civic resistance against the Gorleben salt dome, the new site selection procedure was initiated and the operator BGE eliminated said salt dome in the first phase due to a missing geological overburden. Political opportunism played a major role in determining Gorleben as a site for a repository in Germany. Only by citizens acting as a corrective to this uneven decision-making in the past it was possible that the new site selection process was ever initiated [35,82]. The relevance of this discussion was recently presented by Bell [53] in the context of the Canadian search for a repository site. She concludes that a divide between the treatment of knowledges and knowing individuals according to policy and the actual treatment of knowledge in practice: This demonstrates challenges for justice as recognition as well as procedural justice. The survey has shown first tendencies by assessing that the recognition as emotions does not contribute as strongly to a just site decision as trust in the work of the operator. This is congruent to Roeser's finding that emotional arguments are often framed as irrational either by participants with a different opinion or regulating actors, compared to the 'rationality' of scientific arguments [69].

Even scientific knowledge is never neutral, or free of power relations [51] and a corrective is still necessary. Blindly trusting that science will deliver answers is too short-sighted, as the final decisions for a repository might need to be made by weighing up between similarly well-suited sites. To avoid injustices in such situations, scientificity cannot function as the single most

important determinant in the perception of justice. It can only unfold its full potential along the dimensions of procedure, distribution, and recognition.

*5.3. What Other Research Needs Arise Due to the Constitution of the Sample?*

The constitution of the sample determines the results of this survey and still needs critical examination. Based on procedural observations it can be concluded, that the constitution of the sample is similar to the constitution of the participants of the ongoing site selection process: There is a high degree of participants that have an academic degree, that are older than the average German citizen, and have a certain degree of affection (due to living in a sub-area, past experiences, or spatial proximity to a storage site or nuclear power plant). As already mentioned, there is no full representativity of any kind possible, which limits the results of this survey to the public that is involved in the ongoing process. A general challenge are structural and societal barriers that hinder marginalized people to speak and often result in improper recognition (cf. [50]). From the perspective of justice that was presented in Section 2, this is a challenge for the perception of justice and for a just site selection itself. Aspects of marginalization or unequal treatment of participants have only been generally assessed within the survey but those matters need more detailed, qualitative insights.

In the survey, most participants answered that the inclusion of participants, regardless of their cultural, social, financial, or national background is necessary (cf. RJ1, CQ5). In reality, such idealistic assessments soon reach boundaries: The level of language is non-inclusive. This is not only due to the German language within the process, but also due to the level of scientificity within the discourse and especially, the continuation of the process. Most participants of the survey regularly participate in the ongoing site selection process (some even for their employment) and therefore know to a certain extent what the current state is and what the background to most debates is. This is more difficult for newcomers of the process and will potentially restrict access in the future. This is a vital challenge to notions of justice as recognition, as more people will partake in the process by every step that concretizes the site selection. It can be assumed that the results are applicable to most participants of the ongoing site selection process. In order to enable more general assessments, notions of representativity have to be revisited in future surveys.

## 6. Conclusions

This article thematized the following question: Which aspects of justice are particularly important for the German repository site to be perceived as just? The survey provides empirical insights that partly support notions about justice that are established within the scholarly literature.

The role of a science-informed site decision is regarded as highly important for a strong perception of a just site selection for an HLW repository in Germany. Distributional aspects such as utilitarianism, retribution, or exemption of environmentally burdened areas are generally not approved, but the analysis of the schematic maps regarding land use has shown that participants tend to approve a repository site that is close to an (in)active nuclear power plant or interim storage site. This cannot be interpreted as a retributive argument though, as the survey participants specifically rejected retribution. It can be explained by the role of minimizing transports, thus appealing to risk reduction. Notions of intergenerational recognition are generally assessed as important to the perception of justice, meaning that the interests and possibilities of future generations have to be considered, but people assess differently how this can be achieved, e.g., either by enabling retrievability or by finally closing the repository after the waste is stored.

As the study has shown, perceptions of justice are ambiguous and need to be analyzed against personal stakes and experiences to gain additional insights. Future research also needs to realistically assess notions of recognition, as societal desirability potentially affects the assessment of statements. The monopolistic status of scientificity for a just site selection of a nuclear waste repository needs additional critical examination, especially by consider-

ing personal affectedness, scientific uncertainties, and practices of knowledge generations. As Krütli et al. [17] argued it is necessary that citizens can work as a corrective for uneven or technocratic decision making. The results of the survey have therefore demonstrated that one dimension (or even one single aspect) of justice cannot account for the whole complexity of justice.

Although this study has been carried out in the context of nuclear waste management, especially the repository site selection process in Germany, certain elements are interesting for other fields, e.g., climate change discourse. Wang et al. [83] present a study in which they conclude that emotional responses to climate change topics explain support for new policies. Molek-Kozakowska [84] describes how progress narratives are used in climate change discourse for social mobilization and the role of placing trust in science. Schwarz [85] has shown for a wind energy case study how people with a scientific background (such as professors, or PhDs) have a special role with more credibility in participatory processes and how this bears the potential for outcome dissatisfaction, due to uneven power relations. While those studies present interesting points of departure, the interplay of the role of science and the perception of justice has rarely been thematized. Therefore, the presented study can function as a first inquiry, its hypothesis can be tested in other fields. Of course, this does not only refer to climate change but to other socio-technical challenges (especially ones that comprise new infrastructural adaptation) as well. The fields of environmental and energy justice can potentially benefit from the results of this study by considering how the role of scientific knowledge, and scientific knowledge generation is perceived in relation to justice. This study has shown, that scientific knowledge has a high importance but a critical balance to this monopolistic status is missing.

**Funding:** This research was funded by Federal Ministry for the Environment, Nature Conservation, Building and Nuclear Safety, grant number 02E11849C and Niedersächsisches Vorab der Volkswagenstiftung, grant number 02E11849C.

**Institutional Review Board Statement:** Ethical review and approval were waived for this study due to voluntariness and anonymity of the survey.

**Informed Consent Statement:** Informed consent was obtained from all subjects involved in the study.

**Data Availability Statement:** The data set generated in the study is available from the author upon justified request.

**Acknowledgments:** The publication of this article was funded by Freie Universität Berlin.

**Conflicts of Interest:** The author declares no conflict of interest.

## Notes

[1] Although the Repository Site Selection Act specifies a fixed date for the designation of a repository site (2031), the Federal Company for Radioactive Waste Disposal (BGE) has already communicated that the designation will probably take place between 2046 and 2068.

[2] The term 'generation' is usually used as a comparative term to visualize the abstract notion of time or long time spans. Statistically speaking one generation last approx. 25 years. Additionally, a 'generation' is characterized by similar social imprints. Nonetheless, the term 'generation' is strongly debated, as it does not provide a clear distinction between people living in different times. In this contribution the term 'generation' will be used as a concept of people in the future.

[3] This data is not publically available, but can be requested from the operator BASE.

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
