# Peer review of "Is It All about a Science-Informed Decision? A Quantitative Approach to Three Dimensions of Justice and Their Relation in the Nuclear Waste Repository Siting Process in Germany"

_societies, doi:10.3390/soc12060179_

Round 1

Reviewer 1 Report (New Reviewer)

This is a very interesting paper that examines perceptions of justice in nuclear waste siting in Germany. In the context of climate change (and even current geopolitical concerns), the issue of nuclear energy, and particularly nuclear waste is an important topic (with a very interesting and controversial history in Germany). While many studies of this genre are qualitative, the author uses quantitative methods which provides an interesting methodological slant to studying the topic at hand. Namely, their interest is in examining which aspects of justice are perceived as being most important when thinking about a just site selection process. This paper makes an interesting and relevant contribution to the rich and developing field of energy and environmental justice that examine various tenets of justice as well as the specific niche of energy justice and nuclear waste siting. The paper examines these in detail and provides some interesting and perhaps surprising results that should be made known. The manuscript is presented in a well-structured manner, the methodology and experimental design are for the most part appropriate, the figures and results are for the most part comprehensible, and the conclusions are for the most part consistent with evidence presented.

While I firmly believe this study makes a valuable contribution to the literature, there are some very major revisions that ought to be completed before the manuscript can be accepted. I have made extensive comments detailing these issues within the pdf, and I will summarize the main components here.

1.       Introduction: I think a little more groundwork could be done in the introduction to situate the literature and situate your contribution to the literature as well as the broader significance and implications of this work. Importantly, this also means the broader significance of this work beyond nuclear waste siting (i.e. justice implications more broadly).

2.       Gap in the literature: I have noted this in comments in the pdf, but frequently the author says (1) this hasn’t been studied (e.g., line 202) or (2) it has been studied but in a different context (e.g., line 205). I think the author needs to make these statements more nuanced. Often where they say this hasn’t been studied at all, it has been studied – and they’re missing key citations (e.g. emotions and nuclear energy). And for (2), yes, context is important, but I think the author needs not discount the existing literature just because it studies a different context (then why should we care about Germany, it’s just another context). Yes context is important, but instead a little more work could be done to integrate and value the existing work and situate this work as building from it, contributing to it, and strengthening it, while ALSO providing new insights.

3.       I think Germany has quite a unique history of environmentalism, green movements, and anti-nuclear movements that are curiously missing from the initial discussion. Additionally, Germany was one of the very prominent cases of nuclear phase-out following Fukushima. The author need not dwell on these, but to not mention them (apart from the phase-out itself) is in my view a significant weakness. Some of this context should be examined, especially since the author spends so much time saying that Germany is so unique and this study provides such a unique contribution to the literature.

4.       Similarly, we are missing the German context of nuclear waste siting, who are the key actors, how is the process being done, etc. And we’re not sure if you’re examining Germany as a whole, or one specific sub-area.

5.       In terms of literature, the citations are relevant and for the most part reflect the state of the art, but there are significant gaps. Some are mentioned in pdf comments, but there are some key relevant citations missing:

Perception of procedural justice nuclear waste

Sundqvist, Göran, and Mark Elam. 2010. “Public Involvement Designed to Circumvent Public Concern? The ‘Participatory Turn’ in European Nuclear Activities.” Risk, Hazards & Crisis in Public Policy 1 (4): 203–229.

Bell, MZ. (2020) Spatialising Procedural Justice: Fairness and Local Knowledge Mobilisation in Nuclear Waste Siting. Local Environment, 26:1, 165-180,

Krütli, P., Stauffacher, M., Pedolin, D., Moser, C., & Scholz, R. W. (2012). The Process Matters: Fairness In Repository Siting For Nuclear Waste. Social Justice Research, 25(1), 79-101.

Chilvers, J., & Burgess, J. (2008). Power Relations: The Politics Of Risk And Procedure In Nuclear Waste Governance. Environment and Planning A, 40(8), 1881.

Perception of procedural justice – environmental technologies 

Song, Hwanseok, Hang Lu, and Katherine A. McComas. "The Role of Fairness in Early Characterization of New Technologies: Effects on Selective Exposure and Risk Perception." Risk Analysis 41.9 (2021): 1614-1629.

Perception of recognition justice – nuclear waste

Hurlbert, Margot, and Jeremy Rayner. 2018. “Reconciling Power, Relations, and Processes: The Role of Recognition in the Achievement of Energy Justice for Aboriginal People.” Applied Energy 228: 1320–1327.

Stanley, Anna. 2009b. “Representing the Knowledges of Aboriginal Peoples: The ‘Management’ of Diversity in Canada’s Nuclear Fuel Waste.” In Nuclear Waste Management in Canada: Critical Issues, Critical Perspectives, edited by Darrin Durant and Genevieve Fuji Johnson, 106–129. Vancouver, CA: University of British Columbia Press.

Perceptions of justice in nuclear waste siting

Vilhunen, Tuuli, Matti Kojo, Tapio Litmanen, and Behnam Taebi. 2019. “Perceptions of Justice Influencing Community Acceptance of Spent Nuclear Fuel Disposal. A Case Study in Two Finnish Nuclear Communities.” Journal of Risk Research, 1–24.

New special issue on trust in nuclear waste siting (I’m citing the introduction here, but you should look at the whole special issue if you are going to discuss trust directly).

Lehtonen, Markku, et al. "Introduction to the special issue “Trust, mistrust, distrust, and trust-building in the nuclear sector: historical and comparative experience from Europe”." Journal of Risk Research (2022): 1-15.

Emotions and nuclear waste

Sjöberg, L. Emotions and Risk Perception. Risk Manag 9, 223–237 (2007). https://doi.org/10.1057/palgrave.rm.8250038

6.       I’m not sure the ties to criminology are particularly helpful, I think they detract from a  lot of rich literature in justice in general.

7.       There are some methodological details missing, for example, would like to know more about the sample of survey respondents. Additionally, I’d like to know why sub-areas were chosen instead of all population, this needs to be better justified.

8.       Another methodological issue, in terms of method following the research question, I'm not sure I understand why the control questions were used to identify "the most important aspect of justice". First, why are the control questions the main focus of the results. And second, why not compare composites of distributive, recognition, and procedural justice? These seem to be disjointed and leftover factors. If this isn't the case, it needs to be better explained and justified.

9.       The discussion of each tenet of justice is rich, but each tenet seems to lack a definition, it would be helpful to have some definition as a starting point (rather than just elements of justice, and rather than discussing various threads in a disconnected way.

10.   Line 91, in the discussion of procedural justice as a legitimizing tactic in nuclear waste siting in Europe, Sundqvist and Elam (2010) really key here (see above for citation) and might help.

11.   This is a significant flaw in my opinion. In terms of the results, there is a mismatch between the survey methodology (statements in the survey) and how they are interpreted. Namely, the paper is about justice and perceptions of justice. But many of the measures of justice contain statements not about the most “just” process, but about the “best” process (e.g, page 8, A spatially well-balanced site decision is important for the “best” repository site”). To conflate “just” and “best” process, in my view is a significant problem. The survey was not asking what is the fairest and most just decision, they were being asked which is the best decision, and there is no way of knowing how they interpreted it. The best process to a member of the public might be one that is scientifically sound, safe, lowest risk, lowest exposure, but this may not be the most just process. So I think some reframing of the results and research questions should be done in order to remedy this mismatch. It could be as simple as asking: what aspects of justice are most valued for the best siting process.

12.   In the discussion, the author asks three questions, one of which is: “Can the scientificity of the site selection process be considered as the most relevant determinant of the perception of justice?” I think there is a logical issue here. Specifically, I think the formulation of this question is framed by how the author has interpreted the results (see earlier comment about methodology). To me, the scientific basis of decision-making is not truly an aspect of justice, one that could be included in the context of recognition justice or other elements of justice, not as an “additional factor”. Perhaps this is the novelty of the research and the main contribution of the paper? If the control factors are deliberately selected as being representative of the three tenets of justice, then that’s a separate issue, but it’s unclear how they all relate. In part, I think this stems from a confusion over how the “controls” are implemented in this study, so that is one starting place.

13.   Line 496. As I state in my pdf comment, there are some flaws in the interpretation and discussion of results. The author states that there was a tendency to select sites near to existing nuclear facilities. This is fine. But then to explain this, the author suggests that this could be explained by retributive argument. This *would* also be fine, had the author not directly measured “retribution” and found that survey respondents were not in support of retribution. So it doesn’t make sense to draw on an argument rebutted by the author’s very own data. If this is a misread of the data, then perhaps it could be clarified in the results.

14.   Line 526, 1) it’s hard to say this is the “sole” reason when there is clearly a spectrum of reasons also supported, it might be the most but certainly not the only important factor. 2) Where were the other factors (e.g. procedural factors) in the context of these results?)

15.   Line 565, needs a citation.

16.   Line 580, This is a strongly worded and problematic statement that needs to be cushioned. It certainly IS possible to recognize and speak to marginalized peoples, is it more difficult? It can be, but it's not impossible. Please revise this statement to reflect some nuance.

17.   A little more in the conclusion on the significance and implications of this work would be helpful, especially given the broader context of the journal, why should this be important to a reader not familiar or interested in nuclear waste siting.

18.   Lastly, linguistically, in general, the manuscript requires substantial copy-editing, and often stumbles over itself with language that is unclear. This stems in part from an overuse of passive language where it’s not clear who is saying what and what is being said (see pdf comment, line 34), in part it also stems from some complex sentence formulation and language use that obscures the logic and clarity of the paper.

19. I think the overall focus on science-informed decision-making is a bit too blinkered in my view, there is fascinating data here, and to focus on that is a bit misleading. Of course the "best" decision has to be informed by science, but that in my view doesn't relate strongly to a "just" decision and justice in general, I think justice focuses on the other dimensions here. (similarly epistemic justice cannot in my view be presented by the statement about science - epistemic justice is about diverse knowledges - including science). So this just needs to be cleared up a bit before publication. 

Otherwise, this is a really interesting and important study, that I believe merits publication if the significant issues can be resolved. There is some fascinating and helpful data here. 

Author Response

This is a very interesting paper that examines perceptions of justice in nuclear waste siting in Germany. In the context of climate change (and even current geopolitical concerns), the issue of nuclear energy, and particularly nuclear waste is an important topic (with a very interesting and controversial history in Germany). While many studies of this genre are qualitative, the author uses quantitative methods which provides an interesting methodological slant to studying the topic at hand. Namely, their interest is in examining which aspects of justice are perceived as being most important when thinking about a just site selection process. This paper makes an interesting and relevant contribution to the rich and developing field of energy and environmental justice that examine various tenets of justice as well as the specific niche of energy justice and nuclear waste siting. The paper examines these in detail and provides some interesting and perhaps surprising results that should be made known. The manuscript is presented in a well-structured manner, the methodology and experimental design are for the most part appropriate, the figures and results are for the most part comprehensible, and the conclusions are for the most part consistent with evidence presented. While I firmly believe this study makes a valuable contribution to the literature, there are some very major revisions that ought to be completed before the manuscript can be accepted. I have made extensive comments detailing these issues within the pdf, and I will summarize the main components here.

A: Thank you very much for your detailed review. I was able to incorporate your remarks and thereby improve the overall quality of the contribution. I would also like to thank you for your remarks directly in the PDF version of my manuscript. All of your remarks were helpful and understandable.

  1. Introduction: I think a little more groundwork could be done in the introduction to situate the literature and situate your contribution to the literature as well as the broader significance and implications of this work. Importantly, this also means the broader significance of this work beyond nuclear waste siting (i.e. justice implications more broadly).

A: I improved the introduction. I added a short explanation about how the empirical example of nuclear waste can potentially be transferred to other cases that handle public bads, e.g.  environmental or atmospheric pollution.

  1. Gap in the literature: I have noted this in comments in the pdf, but frequently the author says (1) this hasn’t been studied (e.g., line 202) or (2) it has been studied but in a different context (e.g., line 205). I think the author needs to make these statements more nuanced. Often where they say this hasn’t been studied at all, it has been studied – and they’re missing key citations (e.g. emotions and nuclear energy). And for (2), yes, context is important, but I think the author needs not discount the existing literature just because it studies a different context (then why should we care about Germany, it’s just another context). Yes context is important, but instead a little more work could be done to integrate and value the existing work and situate this work as building from it, contributing to it, and strengthening it, while ALSO providing new insights.

A: Thank you for addressing this aspect. I have now addressed the studies that fit your description and shown why it was important to me to include those into my contribution, e.g. as starting points for my empirical inquiry.

  1. I think Germany has quite a unique history of environmentalism, green movements, and anti-nuclear movements that are curiously missing from the initial discussion. Additionally, Germany was one of the very prominent cases of nuclear phase-out following Fukushima. The author need not dwell on these, but to not mention them (apart from the phase-out itself) is in my view a significant weakness. Some of this context should be examined, especially since the author spends so much time saying that Germany is so unique and this study provides such a unique contribution to the literature.

A: I added a short explanation on the characteristics of the German case at the beginning of the Research chapter, as it provides important background knowledge, but did not properly fit within the introduction that entirely focusses on justice for nuclear waste management.

  1. Similarly, we are missing the German context of nuclear waste siting, who are the key actors, how is the process being done, etc. And we’re not sure if you’re examining Germany as a whole, or one specific sub-area.

A: I added some details on the German process in the beginning of the Research section. In the methods section I added, that the survey was carried out in all of Germany, i.e. that citizens from sub-areas responded as well as citizens that will (probably) not be affected by the site selection process.

  1. In terms of literature, the citations are relevant and for the most part reflect the state of the art, but there are significant gaps. Some are mentioned in pdf comments, but there are some key relevant citations missing:

Perception of procedural justice nuclear waste

Sundqvist, Göran, and Mark Elam. 2010. “Public Involvement Designed to Circumvent Public Concern? The ‘Participatory Turn’ in European Nuclear Activities.” Risk, Hazards & Crisis in Public Policy 1 (4): 203–229.

A: I integrated this publication into the manuscript.

Bell, MZ. (2020) Spatialising Procedural Justice: Fairness and Local Knowledge Mobilisation in Nuclear Waste Siting. Local Environment, 26:1, 165-180.

A: I integrated this publication into the manuscript.

Krütli, P., Stauffacher, M., Pedolin, D., Moser, C., & Scholz, R. W. (2012). The Process Matters: Fairness In Repository Siting For Nuclear Waste. Social Justice Research, 25(1), 79-101.

A: I integrated this publication into the manuscript.

Chilvers, J., & Burgess, J. (2008). Power Relations: The Politics Of Risk And Procedure In Nuclear Waste Governance. Environment and Planning A, 40(8), 1881.

A: I integrated this publication into the manuscript.

Perception of procedural justice – environmental technologies 

Song, Hwanseok, Hang Lu, and Katherine A. McComas. "The Role of Fairness in Early Characterization of New Technologies: Effects on Selective Exposure and Risk Perception." Risk Analysis 41.9 (2021): 1614-1629.

A: I did not integrate this publication into the manuscript, as it is already covered by other publications, e.g. Roeser (2011). Additionally, risk perception is not the primary aim of my inquiry (but provides an interesting point of departure for future publications).

Perception of recognition justice – nuclear waste

Hurlbert, Margot, and Jeremy Rayner. 2018. “Reconciling Power, Relations, and Processes: The Role of Recognition in the Achievement of Energy Justice for Aboriginal People.” Applied Energy 228: 1320–1327.

A: I integrated this publication into the manuscript.

Stanley, Anna. 2009b. “Representing the Knowledges of Aboriginal Peoples: The ‘Management’ of Diversity in Canada’s Nuclear Fuel Waste.” In Nuclear Waste Management in Canada: Critical Issues, Critical Perspectives, edited by Darrin Durant and Genevieve Fuji Johnson, 106–129. Vancouver, CA: University of British Columbia Press.

A: I did not integrate this publication into the manuscript, as I did not have the time to read it (short reviewing times) but already read other well-fitting publications on this topic, which I included, e.g. Nowlin, Conner (2019).

Perceptions of justice in nuclear waste siting

Vilhunen, Tuuli, Matti Kojo, Tapio Litmanen, and Behnam Taebi. 2019. “Perceptions of Justice Influencing Community Acceptance of Spent Nuclear Fuel Disposal. A Case Study in Two Finnish Nuclear Communities.” Journal of Risk Research, 1–24.

A:

New special issue on trust in nuclear waste siting (I’m citing the introduction here, but you should look at the whole special issue if you are going to discuss trust directly).

Lehtonen, Markku, et al. "Introduction to the special issue “Trust, mistrust, distrust, and trust-building in the nuclear sector: historical and comparative experience from Europe”." Journal of Risk Research (2022): 1-15.

A:

Emotions and nuclear waste

Sjöberg, L. Emotions and Risk Perception. Risk Manag 9, 223–237 (2007). https://doi.org/10.1057/palgrave.rm.8250038

A: I integrated this publication into the manuscript.

  1. I’m not sure the ties to criminology are particularly helpful, I think they detract from a lot of rich literature in justice in general.

A: The tie to criminology is helpful as it provides a theoretical background to matters of justice that are relevant to nuclear communities, i.e. that they either already served their share of responsibility or should be used a preferred site for a repository as the waste was generated within their vicinity. This argumentation is omnipresent in the German site selection process but is rarely thematized theoretically. I took away the stress from this concept in the beginning of the paper, so that it becomes apparent, that it is just another aspect of distributive justice.

7.There are some methodological details missing, for example, would like to know more about the sample of survey respondents. Additionally, I’d like to know why sub-areas were chosen instead of all population, this needs to be better justified.

A: I added a clarifying passage on this: “The latter refinement of the German population was chosen as nuclear waste management is a niche topic and due to the simultaneity of global (e.g. environmental pollution or climate change) and national challenges, it was presumed that people how do not live in a sub-area also tend not to participate in the site selection process. This assessment is de-rived from data from the operator BASE: This data shows that there were proportionally more participants at the sub-areas conference from federal states (e.g. Lower Saxony and Bavaria) that contain a large share of sub-areas2.” – The footnote says: “This data is not publically available, but can be requested from the operator BASE.” Additionally, I inserted a map showing the spatial distribution of the respondents against the sub-areas distribution and a short table including the most relevant data that characterizes the sample.

  1. Another methodological issue, in terms of method following the research question, I'm not sure I understand why the control questions were used to identify "the most important aspect of justice". First, why are the control questions the main focus of the results. And second, why not compare composites of distributive, recognition, and procedural justice? These seem to be disjointed and leftover factors. If this isn't the case, it needs to be better explained and justified.

A: Thank you for pointing this out. As I have not used any existing scales to assess the perception of procedure, distribution, and recognition, I did not want to compare those by simply forming a value (as the relation of the assessed factors is unclear). I therefore added the ‘control questions’ as a possibility to compare those aspects in a non-obvious way (without naming them procedural justice, or intergenerational justice). Due to your comment I changed their designation to ‘Comparsion Questions’. This should clarify that they are used to compare between one another. I clarified this in the manuscript.

  1. The discussion of each tenet of justice is rich, but each tenet seems to lack a definition, it would be helpful to have some definition as a starting point (rather than just elements of justice, and rather than discussing various threads in a disconnected way.

A: I added a short working definition of each dimension of justice at the end of each justice chapter to provide a base for the results and discussion.

  1. Line 91, in the discussion of procedural justice as a legitimizing tactic in nuclear waste siting in Europe, Sundqvist and Elam (2010) really key here (see above for citation) and might help.

A: I added Elam, Sundqvist (2011) as an enrichment to this discussion, by showing their insights from the Swedish process.

  1. This is a significant flaw in my opinion. In terms of the results, there is a mismatch between the survey methodology (statements in the survey) and how they are interpreted. Namely, the paper is about justice and perceptions of justice. But many of the measures of justice contain statements not about the most “just” process, but about the “best” process (e.g, page 8, A spatially well-balanced site decision is important for the “best” repository site”). To conflate “just” and “best” process, in my view is a significant problem. The survey was not asking what is the fairest and most just decision, they were being asked which is the best decision, and there is no way of knowing how they interpreted it. The best process to a member of the public might be one that is scientifically sound, safe, lowest risk, lowest exposure, but this may not be the most just process. So I think some reframing of the results and research questions should be done in order to remedy this mismatch. It could be as simple as asking: what aspects of justice are most valued for the best siting process.

A: This error happened due to translation difficulties and has been corrected (in the questionnaire, the statements were accompanied with a question that elaborated on this terminology). I also added a paragraph in the beginning that shortly elaborates on the relation of the best repository site and the most just repository site.

  1. In the discussion, the author asks three questions, one of which is: “Can the scientificity of the site selection process be considered as the most relevant determinant of the perception of justice?” I think there is a logical issue here. Specifically, I think the formulation of this question is framed by how the author has interpreted the results (see earlier comment about methodology). To me, the scientific basis of decision-making is not truly an aspect of justice, one that could be included in the context of recognition justice or other elements of justice, not as an “additional factor”. Perhaps this is the novelty of the research and the main contribution of the paper? If the control factors are deliberately selected as being representative of the three tenets of justice, then that’s a separate issue, but it’s unclear how they all relate. In part, I think this stems from a confusion over how the “controls” are implemented in this study, so that is one starting place.

A: As stated above I clarified your concern over the term ‘control question’. The (now so-called) ‘comparison questions’ stand as representatives of the assessed aspects of the tenets of justice. The assessment that scientificity is the most important contributor to a just repository site is easily understandable, but the quantitative approach does not generate any insights into the reasoning behind this assessment. Therefore, the question should in my opinion not be changed and is highly relevant here – and also a distinguishing characteristic of this contribution. I would therefore not like to change the question, as I think that the correction on the comparison question will dissolve most of your concern. I improved the discussion though, as I think that it helps shed clarity on the problem that this assessment may cause for notions of justice.

  1. Line 496. As I state in my pdf comment, there are some flaws in the interpretation and discussion of results. The author states that there was a tendency to select sites near to existing nuclear facilities. This is fine. But then to explain this, the author suggests that this could be explained by retributive argument. This *would* also be fine, had the author not directly measured “retribution” and found that survey respondents were not in support of retribution. So, it doesn’t make sense to draw on an argument rebutted by the author’s very own data. If this is a misread of the data, then perhaps it could be clarified in the results.

A: This is a highly interesting point as there is an obvious contradiction. If the question is openly formulated, the participants of the survey disagree that retribution should play a role but if the question is formulated in a hidden manner, the participants agree that it is a good idea to build the repository close to nuclear power plants. There are some reasons for this: transportation ways are eliminated and therefore risks can be reduced. It is neglected though, that there are still more sites that store nuclear waste and transports can therefore not be entirely dismissed. I added this explanation to this notion.

  1. Line 526, 1) it’s hard to say this is the “sole” reason when there is clearly a spectrum of reasons also supported, it might be the most but certainly not the only important factor. 2) Where were the other factors (e.g. procedural factors) in the context of these results?)

A: For this question I addressed the scientificity as it was the most strongly approved factor for a just site decision. I changed the phrasing for 1) as well.

  1. Line 565, needs a citation.

A: I added the citation at the end of the phrase (Fricker 2007).

  1. Line 580, This is a strongly worded and problematic statement that needs to be cushioned. It certainly IS possible to recognize and speak to marginalized peoples, is it more difficult? It can be, but it's not impossible. Please revise this statement to reflect some nuance.

A: Thank you for pointing this out. I added some nuance. It now reads: “A general challenge are structural and societal barriers that hinder marginalized people to speak and often result in improper recognition (cf. [48]).”

17.A little more in the conclusion on the significance and implications of this work would be helpful, especially given the broader context of the journal, why should this be important to a reader not familiar or interested in nuclear waste siting.

A: I added a paragraph on this in the concluding section.

18.Lastly, linguistically, in general, the manuscript requires substantial copy-editing, and often stumbles over itself with language that is unclear. This stems in part from an overuse of passive language where it’s not clear who is saying what and what is being said (see pdf comment, line 34), in part it also stems from some complex sentence formulation and language use that obscures the logic and clarity of the paper.

A: Thank you very much for pointing this out. I addressed your criticism during the proof-reading process and simplified statements that were not entirely clear or difficult to read.

  1. I think the overall focus on science-informed decision-making is a bit too blinkered in my view, there is fascinating data here, and to focus on that is a bit misleading. Of course the "best" decision has to be informed by science, but that in my view doesn't relate strongly to a "just" decision and justice in general, I think justice focuses on the other dimensions here. (similarly epistemic justice cannot in my view be presented by the statement about science - epistemic justice is about diverse knowledges - including science). So this just needs to be cleared up a bit before publication. 

A: I addressed this point by clearing up imprecise formulations in the manuscript as well as an explanatory paragraph in the beginning.

Otherwise, this is a really interesting and important study, that I believe merits publication if the significant issues can be resolved. There is some fascinating and helpful data here. 

A: Thank you very much for reviewing my manuscript in such great details and with helpful comments.

Reviewer 2 Report (New Reviewer)

The work raises the need to determine a scientific method when locating a space to deposit nuclear waste, taking into account the principles of environmental justice.

 The method is based on the use of a survey, perhaps with too few informants, which can reduce the statistical representativeness of the results.

The results obtained are novel and the discussion is very rich.

The bibliography used is relevant to the subject of study and includes the main and most recent works on the subject analyzed.

The work can be published without the need to introduce relevant changes

Author Response

Thank you for reviewing my manuscript.

Round 2

Reviewer 1 Report (New Reviewer)

Thank for thoroughly addressing my concerns in highlighting weaknesses of the paper. The revised manuscript is much improved, engages more deeply with the literature, explains its methodological approach, and is more logical and coherent. 

There is only one remaining issue, rather small but quite important.  While the paper does engage with the literature thoroughly in the literature review, it is still missing a couple of sentences in the introduction and conclusion which situate the paper in the context of existing literature and explains its significance and contribution to those literatures, that is quite important and yet it is missing. 

(In relation to comment 1, the author addressed the last half of my comment by "adding significance of this work beyond nuclear waste siting”, but they did not address the first half of my comment, which was to “to situate the literature and situate your contribution to the literature as well as the broader significance and implications of this work.”. The author needs to provide a brief overview of what literatures they are speaking to, and how their work makes a contribution to those literatures? Is it energy justice? Environmental justice? Philosophy? What specifics literature are you speaking to in your paper? You need to demonstrate a and position yourself in the literature in the introduction and then flesh this out in the literature review.) 

Once this is done, the paper in my view will be complete and ready for publication. It's a great study! 

Author Response

Reviewer: Thank for thoroughly addressing my concerns in highlighting weaknesses of the paper. The revised manuscript is much improved, engages more deeply with the literature, explains its methodological approach, and is more logical and coherent.

There is only one remaining issue, rather small but quite important.  While the paper does engage with the literature thoroughly in the literature review, it is still missing a couple of sentences in the introduction and conclusion which situate the paper in the context of existing literature and explains its significance and contribution to those literatures, that is quite important and yet it is missing.

(In relation to comment 1, the author addressed the last half of my comment by "adding significance of this work beyond nuclear waste siting”, but they did not address the first half of my comment, which was to “to situate the literature and situate your contribution to the literature as well as the broader significance and implications of this work.”. The author needs to provide a brief overview of what literatures they are speaking to, and how their work makes a contribution to those literatures? Is it energy justice? Environmental justice? Philosophy? What specific literature are you speaking to in your paper? You need to demonstrate and position yourself in the literature in the introduction and then flesh this out in the literature review.)

Once this is done, the paper in my view will be complete and ready for publication. It's a great study!

Author: Thank you very much for your positive feedback. I addressed your last comment the following way: My study is situated within the broad field of justice research, but mainly environmental justice and energy justice (as both apply the same framework and nuclear waste is a topic at the intersection of both domains). It now reads: “By employing this approach, this study is situated in the field of environmental and energy justice. As authors within both domains have qualitatively contributed a wide range of considerable aspects that are relevant to justice, this study provides a per-spective how justice research about a so-called public bad [11] can be carried out quantitatively and what implications require special emphasis. Additionally, this study contributes to the existing literature by testing whether certain aspects that have been discussed theoretically (e.g. the role of retributive argumentation in justice) are perceived by survey respondents.” (This section is now part of the introduction). Additionally, I added some insights by the end of the manuscript, showing how environmental and energy justice can benefit from the insights of this study.

Round 3

Reviewer 1 Report (New Reviewer)

Thank you for addressing my final comments, I believe it is now ready to be published, congratulations! 

This manuscript is a resubmission of an earlier submission. The following is a list of the peer review reports and author responses from that submission.

Round 1

Reviewer 1 Report

Thank you for the opportunity to review the manuscript titled “Perceptions of justice in the search for a nuclear waste repository in Germany”. 

The paper focuses “around the question of how – albeit all injustices that are inherent to the deployment of nuclear energy – people perceive justice when dealing with nuclear waste” (66-68). The title and the research question suggest that the opinions of the Germans (people) are the subject of the research.

The case of nuclear waste management in Germany is very interesting because of strong anti-nuclear movement in the country and the political phase-out decisions regarding nuclear power. I also totally agree with the authors that “justice is (…) inherent in discussions about nuclear waste, its management, and the safety of disposal” (49). There surely is a demand for the analysis of German perceptions of justice concerning nuclear waste management. The starting points of the manuscript are very promising but the discrepancy between the research question and the (non-probability) sample of the survey is the problem. The title of the manuscript also suggests that there is survey data to be analyzed that represents the opinions of the Germans, but this is not the case. “Participants of the survey were spread all over Germany etc” (230) but the respondents don’t represent the Germans according to their place of residence and apparently not in any other way. In the paper the authors openly describe how “a Germany-wide survey” data was collected but in Section 3 the authors don’t state that data would represent the views of the Germans. My question and concern is: which group of respondents does the sample represent? What is the target population of the survey?

The authors discuss “the constitution of the sample” and they note that “The results of this survey need to be analyzed in a categorized manner, dividing the sample based on their experiences with nuclear waste, as well as their spatial proximity to sites of nuclearity (such as nuclear power plants or nuclear (interim) storage sites), and therefore their stake in the site selection process. This will enable more precise assessments of justice based on individual and/or social characteristics.” (546-550) Why wasn’t this done? As the process roles of the respondents were asked (see Table 1) it could have provided an interesting opportunity to look at differences of opinions between various groups.

Author Response

Reviewer: Thank you for the opportunity to review the manuscript titled “Perceptions of justice in the search for a nuclear waste repository in Germany”. The paper focuses “around the question of how – albeit all injustices that are inherent to the deployment of nuclear energy – people perceive justice when dealing with nuclear waste” (66-68). The title and the research question suggest that the opinions of the Germans (people) are the subject of the research.

Author: Thank you very much for reviewing my manuscript. As I understand your criticism regarding the suggestion of the title, I slightly changed the title: “Perceptions of justice in the German site selection process for a nuclear waste repository”. The research question has been improved to actually reflect the precise question that led my research: “Which aspects of justice are of particular importance for the German repository site to be perceived as just?”

R: The case of nuclear waste management in Germany is very interesting because of strong anti-nuclear movement in the country and the political phase-out decisions regarding nuclear power. I also totally agree with the authors that “justice is (…) inherent in discussions about nuclear waste, its management, and the safety of disposal” (49). There surely is a demand for the analysis of German perceptions of justice concerning nuclear waste management. The starting points of the manuscript are very promising but the discrepancy between the research question and the (non-probability) sample of the survey is the problem. The title of the manuscript also suggests that there is survey data to be analyzed that represents the opinions of the Germans, but this is not the case. “Participants of the survey were spread all over Germany etc” (230) but the respondents don’t represent the Germans according to their place of residence and apparently not in any other way. In the paper the authors openly describe how “a Germany-wide survey” data was collected but in Section 3 the authors don’t state that data would represent the views of the Germans. My question and concern is: which group of respondents does the sample represent? What is the target population of the survey?

A: The question of representativity has been addressed in the sample now and the sample is now described within an own section (section 4) to clarify its constitution.

R: The authors discuss “the constitution of the sample” and they note that “The results of this survey need to be analyzed in a categorized manner, dividing the sample based on their experiences with nuclear waste, as well as their spatial proximity to sites of nuclearity (such as nuclear power plants or nuclear (interim) storage sites), and therefore their stake in the site selection process. This will enable more precise assessments of justice based on individual and/or social characteristics.” (546-550) Why wasn’t this done? As the process roles of the respondents were asked (see Table 1) it could have provided an interesting opportunity to look at differences of opinions between various groups.

A: You are entirely right, personally I already conducted this analysis. For this general manuscript this would have been too much and would have shifted the focus away from the actual research question. Right now, I am preparing a manuscript that will focus entirely on how different groups/clusters perceive what is just for nuclear waste management.

Reviewer 2 Report

This manuscript presents results of an extensive survey concerning the justice-related aspects of high-level radioactive waste management, carried out among German citizens and experts. The survey was certainly carefully conducted, although the manuscript does not provide full details, such as the specific questions posed. The topic is interesting and highly relevant for the journal, and topical for the current nuclear waste discussions in Germany and beyond. However, as the manuscript stands now, it has a number of considerable weaknesses, and I could only recommend its publication in a thoroughly revised form. “Revise and resubmit“ would probably best describe my judgement.

General remarks

The first major problem concerns the language. At many instances, I failed to understand the meaning of the sentences, even after a couple of readings. This is partly purely a question of English language, which should be thoroughly revised, preferably by a native speaker. However, my problems of understanding stem largely also from the lack of definition of key concepts, as well as an excessively complicated writing style.

Overall, I have the impression that there is plenty of material and lots of possible points of interest in the paper, but as it stands now, the paper is not sufficiently well focused. In other words, perhaps there’s just too much “going on“ in the paper now, and choices would have to be made, to give the paper a clearer focus and line of argumentation. This problem may well stem from the lack of a sufficiently clearly focused research question. At the end of the introduction, the manuscript states: “This article revolves around the question of how – albeit all injustices that are inherent to the deployment of nuclear energy – people perceive justice when dealing with nuclear waste.“ This is fine as a general aim, but a better focused question would be needed. At the minimum, the question should be reformulated to apply to Germans, and not “people“ in general. By the way, it is not a good idea to use an expression such as “revolves around“ when describing the research question. Must be precise and not “revolve around“ an issue…

Disconnect between the theoretical and empirical sections

Perhaps the single most serious shortcoming is the disconnect between the theoretical part, i.e., the literature survey, and the empirical analysis. The literature survey should, in my view, end in the presentation of the conceptual framework applied in this paper. Essentially, this would imply describing the dimensions of justice that were applied in the survey – and justifying the conceptual choices with reference to earlier literature. This should then lead, probably best presented in the methodology section, of the ways in which those concepts were operationalised in the survey. In essence, what questions did you pose in the survey, under each of your justice category? How did you define and operationalise, through concrete survey questions, notions such as transparency, comprehensibility, participation, honesty, power, utilitarianism, retribution, compensatory justice, scientificity, or exemption of environmentally burdened regions? It is very difficult to follow the line of argumentation in the paper, largely because these key concepts are not adequately defined. Some concepts, such as retribution, are indeed defined, but at a late stage. The definitions should be provided early on – not mixed with the presentation of results. Defining the concept of scientificity would probably need to be made with particular care, given that it belongs to the group of those problematic, controversial and ambiguous terms such rationality, objectivity, neutrality, etc.

Methods

The main thing lacking in this section is what I alluded to above: description of how the key concepts, that is, justice dimensions analysed in this study, were operationalised through specific survey questions. The rest of the description concerning the survey methods, data, and sample seem fine. But as I mentioned above, the methods section should describe how, in practice, you explored the views of the respondents on the importance of, for example, transparency, honesty, utilitarianism, etc.

You should probably briefly explain the role of the different tests and methods that you mention. No need to go into great detail, but it would be good if you explained things like:

  • What do you mean by the fit of the proposed models? What does the term fit stand for here? In other words, what type of criteria does the Goodness-of-Fit test apply for judging the fit?
  • You say that the usability of the pseudo r2 value is contested, and that you therefore used the Nagelkerke value. Fine, but what was the Nagelkerke value used for? If the usability of the pseudo r2 value is contested, does this apply regardless of use? If it doesn’t, in other words, if the pseudo r2 value can be useful in some situations but not in others, why was it not useful in your case?

Discussion and conclusions

As I mentioned, the entire text would have to be carefully revised, to make sure that it is understandable, not only when it comes to grammatical correctness and syntax, but above all by making sure that the concepts are made clear to the reader early on. I found both the results and discussion sections hard to follow. However, this was particularly frustrating with the discussion section, because seemingly there are a number of potentially interesting findings and conclusions there, but for the most part, they remained somehow almost mysterious and ambiguous to me.

At times, I could not understand the meaning of a specific finding, while at other times I could not see where the findings sprang from, that is, how the results warranted the conclusions made. An example of the latter is the observation – central to the paper – that “The results of the survey have shown that no dimension of justice can be dismissed in favor of another.“ The same message is repeated in the conclusions: It was shown that no dimension of justice can be dismissed in favor of another dimension and even if one aspect of a dimension of justice cannot be achieved still other aspects of this dimension of justice are at play.“ Fine, but on which grounds do you say so? Is the expression “no dimension of justice can be dismissed“ the appropriate one? Why “cannot“? What happens if one dimension is dismissed? Should you rather say that the respondents did not see the trade-offs between the dimensions in the same way as some/many scholars have done? I understand that you took issue with the trade-offs thinking, but did not fully grasp what the issue was. Maybe this is a topic you might pursue further, when revising and better focusing your conclusions?

You will find further, more detailed, comments inserted directly into the attached pdf manuscript file. They are not an exhaustive list of comments, but rather illustrative. The entire paper should be rewritten, following hopefully the suggestions and observations I have made above.

Author Response

R: This manuscript presents results of an extensive survey concerning the justice-related aspects of high-level radioactive waste management, carried out among German citizens and experts. The survey was certainly carefully conducted, although the manuscript does not provide full details, such as the specific questions posed. The topic is interesting and highly relevant for the journal, and topical for the current nuclear waste discussions in Germany and beyond. However, as the manuscript stands now, it has a number of considerable weaknesses, and I could only recommend its publication in a thoroughly revised form. “Revise and resubmit“ would probably best describe my judgement.

A: Thank you very much for your helpful review. I was able to address your comments in multiple ways. The detailed comments and how I implemented them in the manuscript are listed below.

General remarks

R: The first major problem concerns the language. At many instances, I failed to understand the meaning of the sentences, even after a couple of readings. This is partly purely a question of English language, which should be thoroughly revised, preferably by a native speaker. However, my problems of understanding stem largely also from the lack of definition of key concepts, as well as an excessively complicated writing style.

A: I checked the whole manuscript again and shortened sentences that were too long to understand. Additionally, a second person checked the language of the manuscript and the language quality should now be greatly improved. The key concepts are now explained, additionally I provided a detailed table of how the aspects of justice were operationalized for the study.

R: Overall, I have the impression that there is plenty of material and lots of possible points of interest in the paper, but as it stands now, the paper is not sufficiently well focused. In other words, perhaps there’s just too much “going on“ in the paper now, and choices would have to be made, to give the paper a clearer focus and line of argumentation. This problem may well stem from the lack of a sufficiently clearly focused research question. At the end of the introduction, the manuscript states: “This article revolves around the question of how – albeit all injustices that are inherent to the deployment of nuclear energy – people perceive justice when dealing with nuclear waste.“ This is fine as a general aim, but a better focused question would be needed. At the minimum, the question should be reformulated to apply to Germans, and not “people“ in general. By the way, it is not a good idea to use an expression such as “revolves around“ when describing the research question. Must be precise and not “revolve around“ an issue…

A: I reformulated the research question so that it actually represents the RQ that led my research. Thereby the paper got a clearer focus, as I also addressed the comments you provided in the PDF document. I narrowed my selection of dimensions down to the three tenets of distribution, procedure and recognition.

Disconnect between the theoretical and empirical sections

R: Perhaps the single most serious shortcoming is the disconnect between the theoretical part, i.e., the literature survey, and the empirical analysis. The literature survey should, in my view, end in the presentation of the conceptual framework applied in this paper. Essentially, this would imply describing the dimensions of justice that were applied in the survey – and justifying the conceptual choices with reference to earlier literature. This should then lead, probably best presented in the methodology section, of the ways in which those concepts were operationalised in the survey. In essence, what questions did you pose in the survey, under each of your justice category? How did you define and operationalise, through concrete survey questions, notions such as transparency, comprehensibility, participation, honesty, power, utilitarianism, retribution, compensatory justice, scientificity, or exemption of environmentally burdened regions? It is very difficult to follow the line of argumentation in the paper, largely because these key concepts are not adequately defined. Some concepts, such as retribution, are indeed defined, but at a late stage. The definitions should be provided early on – not mixed with the presentation of results. Defining the concept of scientificity would probably need to be made with particular care, given that it belongs to the group of those problematic, controversial and ambiguous terms such rationality, objectivity, neutrality, etc.

A: I improved this part by restructuring the whole section about the state of research and the section about the methodological framework. The connection between those two sections should now be clearer. To increase comprehensibility I added an extensive table of how all aspects were operationalized for the survey.

Methods

R: The main thing lacking in this section is what I alluded to above: description of how the key concepts, that is, justice dimensions analysed in this study, were operationalised through specific survey questions. The rest of the description concerning the survey methods, data, and sample seem fine. But as I mentioned above, the methods section should describe how, in practice, you explored the views of the respondents on the importance of, for example, transparency, honesty, utilitarianism, etc.

A: As mentioned before I provided an extensive table concerning all the aspects involved in the survey. Additionally, in the results section I linked each aspect per ID to the table, which enhances comprehensibility.

R: You should probably briefly explain the role of the different tests and methods that you mention. No need to go into great detail, but it would be good if you explained things like:

  • What do you mean by the fit of the proposed models? What does the term “fit“ stand for here? In other words, what type of criteria does the Goodness-of-Fit test apply for judging the fit?
  • You say that the usability of the pseudo r2 value is contested, and that you therefore used the Nagelkerke value. Fine, but what was the Nagelkerke value used for? If the usability of the pseudo r2 value is contested, does this apply regardless of use? If it doesn’t, in other words, if the pseudo r2 value can be useful in some situations but not in others, why was it not useful in your case?

A: I improved the explanations about the employed statistical methods. The Nagelkerke value is a possibility to calculate the pseudo r2. This is now contained within the methodological framework section. For the Goodness-of-Fit model I included some explanation.

Discussion and conclusions

R: As I mentioned, the entire text would have to be carefully revised, to make sure that it is understandable, not only when it comes to grammatical correctness and syntax, but above all by making sure that the concepts are made clear to the reader early on. I found both the results and discussion sections hard to follow. However, this was particularly frustrating with the discussion section, because seemingly there are a number of potentially interesting findings and conclusions there, but for the most part, they remained somehow almost mysterious and ambiguous to me.

A: I restructured and refocused the discussion to synchronize it with the before presented results. It is now also clearer as I improved the research question before.  

R: At times, I could not understand the meaning of a specific finding, while at other times I could not see where the findings sprang from, that is, how the results warranted the conclusions made. An example of the latter is the observation – central to the paper – that “The results of the survey have shown that no dimension of justice can be dismissed in favor of another.“ The same message is repeated in the conclusions: “It was shown that no dimension of justice can be dismissed in favor of another dimension and even if one aspect of a dimension of justice cannot be achieved still other aspects of this dimension of justice are at play.“ Fine, but on which grounds do you say so? Is the expression “no dimension of justice can be dismissed“ the appropriate one? Why “cannot“? What happens if one dimension is dismissed? Should you rather say that the respondents did not see the trade-offs between the dimensions in the same way as some/many scholars have done? I understand that you took issue with the trade-offs thinking, but did not fully grasp what the issue was. Maybe this is a topic you might pursue further, when revising and better focusing your conclusions?

A: As provided per PDF document I addressed the sections that were unclear and improved the formulations. I also edited formulations that were not understandable or wrongly positioned. Especially, the notion about dismissing another dimension of justice in favor of one dimensions of justice has been clarified and improved within the discussion and results chapter.

R: You will find further, more detailed, comments inserted directly into the attached pdf manuscript file. They are not an exhaustive list of comments, but rather illustrative. The entire paper should be rewritten, following hopefully the suggestions and observations I have made above.

A: Thank you very much for providing such detailed comments on my manuscript. I addressed your concerns and improved formulations as well as clarifying the marked sections.

Reviewer 3 Report

This is an interesting manuscript about of justice in the search for a nuclear waste repository in Germany.

It is well referenced and documented, but the results exposition part should be improved.

Missing a table or similar with the questions asked in the survey, since these are diluted in the tangle of statistical data. One could also try to represent the answers of the survey in a more understandable way, since the graphs used are not very clear, and each question presents a different one.

It would not be difficult to briefly explain the NIMBY effect, or at least describe the acronym.

Finally, the conclusions should be based on the results and the statistical analysis carried out.

With a few improvements, I think the work could be published.

Author Response

R: This is an interesting manuscript about of justice in the search for a nuclear waste repository in Germany. It is well referenced and documented, but the results exposition part should be improved.

A: Thank you very much for reviewing my manuscript. I restructured almost all sections of the paper, including the state of research, methodological framework, sample, results, discussion, and conclusion. Therefore, the relation to the research question and all other sections has been strengthened throughout the manuscript.

R: Missing a table or similar with the questions asked in the survey, since these are diluted in the tangle of statistical data. One could also try to represent the answers of the survey in a more understandable way, since the graphs used are not very clear, and each question presents a different one.

A: I added an extensive table containing my whole methodological framework in connection to the state of research. Additionally, I added an ID that I mentioned within the results section to improve comprehensibility and connect each statement and result with the corresponding item of the methodological framework.

R: It would not be difficult to briefly explain the NIMBY effect, or at least describe the acronym.

A: I added the acronym and provided a short explanation (corresponding to the nuclear waste context).

R: Finally, the conclusions should be based on the results and the statistical analysis carried out. With a few improvements, I think the work could be published.

A: As mentioned above, the whole manuscript has been restructured, therefore, the conclusion has been improved as well to address the research question and rely on the results.

Round 2

Reviewer 2 Report

I’m afraid I see very little improvement in the revised manuscript, and a number of new problems that appeared with the revision. My recommendation this time would be “thoroughly and carefully revise and resubmit“. If indeed the author decides to resubmit, I would underline that what is needed is thorough revision, with the correction of the language as the very first task. Let a native speaker go through the text and ask him/her also to check that the sentences and line of argumentation are clear. A second general recommendation would be to greatly simplify and condense the theoretical/literature part of the manuscript, and instead build on the potential key strength of the paper, that is, the empirical survey. The current version of the manuscript tries to achieve far too many things at the same time – you would need a dozen of papers to adequately cover all the terrain that you seem to suggest covering in a single article. One option might be to simply cut the number of survey questions that you seek to cover in this one single paper, and concentrate instead on a sub-set of questions.

In the following, I will first go through my comments from the first round and then continue with additional remarks.

I wrote: The first major problem concerns the language. At many instances, I failed to understand the meaning of the sentences, even after a couple of readings. This is partly purely a question of English language, which should be thoroughly revised, preferably by a native speaker. However, my problems of understanding stem largely also from the lack of definition of key concepts, as well as an excessively complicated writing style.

It is clear that the article has not been re-read by a native speaker – maybe the journal did not allow sufficient time to allow this. Also, the general problems of unclear expressions and undefined or poorly defined concepts still persist to a large extent. I do appreciate the authors effort at clarifying the approach by introducing table 2, which indeed helps the reader to better understand the concepts and reasoning. However, the language, style, and structure of argumentation remain serious problems that need to be corrected before anything else.

I wrote: Overall, I have the impression that there is plenty of material and lots of possible points of interest in the paper, but as it stands now, the paper is not sufficiently well focused. In other words, perhaps there’s just too much “going on“ in the paper now, and choices would have to be made, to give the paper a clearer focus and line of argumentation. This problem may well stem from the lack of a sufficiently clearly focused research question. At the end of the introduction, the manuscript states: “This article revolves around the question of how – albeit all injustices that are inherent to the deployment of nuclear energy – people perceive justice when dealing with nuclear waste.“ This is fine as a general aim, but a better focused question would be needed. At the minimum, the question should be reformulated to apply to Germans, and not “people“ in general. By the way, it is not a good idea to use an expression such as “revolves around“ when describing the research question. Must be precise and not “revolve around“ an issue…

The author narrowed my selection of dimensions down to distributive, procedural and recognition justice, which is helpful as such, but there are still far too many elements, dimensions, and aspects that the paper touches upon, but often without doing this in a sufficiently detailed manner. Table 2 indeed is an illustration of the multiplicity of dimensions, aspects of justice, and specific survey questions, which makes it still very difficult to identify the key messages that the paper seeks to convey.

I wrote: Perhaps the single most serious shortcoming is the disconnect between the theoretical part, i.e., the literature survey, and the empirical analysis. The literature survey should, in my view, end in the presentation of the conceptual framework applied in this paper. Essentially, this would imply describing the dimensions of justice that were applied in the survey – and justifying the conceptual choices with reference to earlier literature. This should then lead, probably best presented in the methodology section, of the ways in which those concepts were operationalised in the survey. In essence, what questions did you pose in the survey, under each of your justice category? How did you define and operationalise, through concrete survey questions, notions such as transparency, comprehensibility, participation, honesty, power, utilitarianism, retribution, compensatory justice, scientificity, or exemption of environmentally burdened regions? It is very difficult to follow the line of argumentation in the paper, largely because these key concepts are not adequately defined. Some concepts, such as retribution, are indeed defined, but at a late stage. The definitions should be provided early on – not mixed with the presentation of results. Defining the concept of scientificity would probably need to be made with particular care, given that it belongs to the group of those problematic, controversial and ambiguous terms such rationality, objectivity, neutrality, etc.

As said above the new table is helpful in that it presents the key concepts in a slightly more concise manner. However, the theory section – introduction and state of research – continue to suffer from largely the same problems as earlier. There are too many concepts and ideas, references to literature, often just one-sentence statements, whose connection to the actual purposes of the article remain unclear. The theory / literature review section should be clearly focused so that it directly serves the specific research objectives of this article. I would recommend that the author cut down the length of section 2 and focus the section with the research task in mind. At the moment, the section tries to cover far too much theoretical and conceptual ground – as does in fact the whole manuscript. Choices need to be made in order to identify the specific contribution that the paper seeks to make to the exiting scholarship. This is needed because the literature on justice is vast, and there is quite a body of literature even on justice issues related to radioactive waste management.

This recently published open access article on ethics and justice relating to radioactive management might be useful especially when revising the literature section. It provides both a host of literature sources and a possibly helpful example of how you might think of presenting the earlier literature and your conceptual framework:

Kojo, M., Vilhunen, T., Kari, M., Litmanen, T. & Lehtonen, M. 2022. The Art of Being Ethical and Responsible: Print Media Debate on Final Disposal of Spent Nuclear Fuel in Finland and Sweden. Social Justice Research 35, 157–187. https://doi.org/10.1007/s11211-022-00391-6

I should return here also to my first-round comment about scientificity. The author has replaced the term by expressions such as scientific decision-making, and science-based decision-making. Unfortunately, I don’t think this helps much. These concepts are vague, disputed, and controversial – hundreds of articles and books have been written on the topics. The author should define the terms, or perhaps better, choose completely different terms to describe what these concepts denote in this paper. For example, scientific decision-making sounds like an oxymoron to me – or rather, scientific decision-making might refer to decisions that scientists make at various stages of the research process. Science-based decision-making is probably closer to what the author refers to here, but even that term leads to discussions about the respective pros and cons of science-based and science-informed decision-making, for example. In the same vein, the way in which the author uses the concept of rationality is questionable in that it seems to equate rationality with something like “decisions should be made by experts/scientists rather than by laymen or politicians“. Again, the literature on rationality is vast, and reducing rationality to such a simple unidimensional definition seems inappropriate to me. There is a fair amount of consensus behind the idea that rationalities are multiple – and that one of them might be called something like “scientific rationality“. However, to avoid complications, my advice would be to avoid using the term altogether. I don’t think it adds anything to the argument in this paper.

Methods

I wrote: The main thing lacking in this section is what I alluded to above: description of how the key concepts, that is, justice dimensions analysed in this study, were operationalised through specific survey questions. The rest of the description concerning the survey methods, data, and sample seem fine. But as I mentioned above, the methods section should describe how, in practice, you explored the views of the respondents on the importance of, for example, transparency, honesty, utilitarianism, etc.

Again, I appreciate the table provided by the author. It is very helpful, indeed. For clarity, I would recommend replacing the expression “conceptualisation“ to something more simple, perhaps just “the survey question“. Moreover, the text should refer to the table and briefly mention/explain its rationale.

I wrote: You should probably briefly explain the role of the different tests and methods that you mention. No need to go into great detail, but it would be good if you explained things like:

  • What do you mean by the fit of the proposed models? What does the term “fit“ stand for here? In other words, what type of criteria does the Goodness-of-Fit test apply for judging the fit?
  • You say that the usability of the pseudo r2 value is contested, and that you therefore used the Nagelkerke value. Fine, but what was the Nagelkerke value used for? If the usability of the pseudo r2 value is contested, does this apply regardless of use? If it doesn’t, in other words, if the pseudo r2 value can be useful in some situations but not in others, why was it not useful in your case?

These issues are now described in somewhat better way in the new version, but problems remain. As for the goodness-of-fit test, the sentence referring to the assessment of the explanatory value etc. is mysterious and unclear. As for the Nagelkerke score, my question from above remains unanswered: if the “general usability“ of this method is “contested“, why did you use it? Was your case somehow different from the “general“ situation in which the usability is contested?

I wrote: As I mentioned, the entire text would have to be carefully revised, to make sure that it is understandable, not only when it comes to grammatical correctness and syntax, but above all by making sure that the concepts are made clear to the reader early on. I found both the results and discussion sections hard to follow. However, this was particularly frustrating with the discussion section, because seemingly there are a number of potentially interesting findings and conclusions there, but for the most part, they remained somehow almost mysterious and ambiguous to me.

I’m sad to say that I found very little improvement in the new version. On every page, there are lots of sentences that are unclear, contain sweeping statements that are not clearly backed up by evidence, and which raise more questions than they answer.

I wrote: At times, I could not understand the meaning of a specific finding, while at other times I could not see where the findings sprang from, that is, how the results warranted the conclusions made. An example of the latter is the observation – central to the paper – that “The results of the survey have shown that no dimension of justice can be dismissed in favor of another.“ The same message is repeated in the conclusions: “It was shown that no dimension of justice can be dismissed in favor of another dimension and even if one aspect of a dimension of justice cannot be achieved still other aspects of this dimension of justice are at play.“ Fine, but on which grounds do you say so? Is the expression “no dimension of justice can be dismissed“ the appropriate one? Why “cannot“? What happens if one dimension is dismissed? Should you rather say that the respondents did not see the trade-offs between the dimensions in the same way as some/many scholars have done? I understand that you took issue with the trade-offs thinking, but did not fully grasp what the issue was. Maybe this is a topic you might pursue further, when revising and better focusing your conclusions?

I must repeat what I said above, on the previous point: I am still rather confused and uncertain about the actual contribution of the paper. A lot of this confusion has to do with the grammar and style of writing, including imprecise use of concepts. The essential first step would be to have someone critically go through the text and check that at least each sentence makes sense, and to minimise ambiguity. Beyond that, plenty of work remains to ensure that the text is also “in scholarly terms“ adequate.

Without going into all the details, in the following, a few additional remarks on specific points in the revised manuscript. These are meant to be examples that serve as a tentative guide when revising the article. This is therefore only a sample - and by no means a comprehensive list of all sentences and passages that I found problematic in this paper.

In the abstract, instead of using passive voice – “X is considered highly important“ etc. –  start the presentation of the results, in the middle of the abstract, by something like “The results showed that the role of a…“

As for the survey questions, many of them seemed a bit self-evident to answer. Most people would certainly agree that just about all of the criteria mentioned in the questions are important. I think it would have been much more instructive to ask the respondents, from the very beginning, to prioritise between different justice-related objectives and criteria. Also, at times I wondered whether the survey questions inquired about the respondents’ views on a just site selection process or simply on what they thought would be in general good site selection process.

In the introduction:

It is probably safer to say that nuclear energy has in Germany been contested especially since the 70s, and not “always“.

Why can nuclear waste be considered an “eternal burden“ only for Germany? Isn’t this true for all countries producing nuclear power?

“Especially albeit all injustices that are inherent to the deployment of nuclear energy“ – what do you mean here?

“…newly-initiated German repository site selection process that intensively deals with justice…“. No, it is your paper that seeks to intensively engage with this topic – not the site selection process.

What do you mean by the term “thematize“? What exactly do you do when you thematize?

State of Research:

adjacent fields that are transferable“ – transferable to what? And from where?

for nuclear waste repositories an equal distribution of the eternal burden is excluded for safety reasons“ – What does this mean???

equal distribution, in the sense of Rawls [21], cannot be achieved“ – what is this sense?

The established NIMBY phenomenon (Not in My Backyard) is theoretically applicable 146 here but requires empirical examination.“ – What is NIMBY, why and how is it “theoretically applicable“, and why does it require empirical examination?

Utilitarianism: I think the concept is in a strange place in this paper. Utilitarianism is an umbrella concept – a normative ethical theory – whereas you have reduced it to a single, specific aspect of distributive justice alongside things like compensation and environmental burden. I find this a bit strange.

the societal handling of such has not yet been sufficiently investigated empirically“ – What does this mean???

Another adjacent factor that is repeatedly mentioned but has not been analyzed more deeply is the influence of emotions on the perception of justice.“ – I’m not sure whether this is true. Perhaps more importantly, I think you should keep in mind that your analysis does not analyse this aspect – and none of the many aspects included in your survey – in any deeper manner either. Your survey touches upon a large number of topics, but does not delve deeper into any of them. This is fine as such, but simply something to keep in mind when writing.

You repeat several times that an empirical approach is necessary. This is fine, but you should perhaps be careful with your claims about what you aim to and can achieve. An empirical survey has its own virtues and weaknesses, as does for example an empirical case study that employs qualitative or mixed methods.

Methods section

To make the diverse state of research on justice in dealing with nuclear waste tangible, a quantitative survey was conducted“. – No, I don’t think you make earlier research tangible, but rather, you operationalise your own key concepts and thus make them tangible, perhaps.

Due to its low participation hurdle, the survey thus represents a baseline survey.“ – What does this mean?

participants did not require any prior knowledge about nuclear energy“ – I think you mean to say that no prior knowledge was required from the participants.

The target population of the survey is not clear“ -  I think you mean the target population for the site selection, or something similar. For your survey, the target population was hopefully clear.

Both tables in the methods section are very helpful. On the classification of questions under different dimensions, relevant aspects of justice, etc., I might have plenty of questions and doubts, though. But well, any categorisation is bound to have its own problems. Maybe the main challenge now is to ensure that your theory section adequately justifies the choices and categorisations made. I am not quite certain that this is the case. The connection between the textual description and the table should probably be made clearer.

You mention the inevitable overlaps between procedural and recognition justice. However, in the table, these overlaps seem problematic. Why is for instance “process length“ under the category of recognition justice and not procedural justice? Likewise, “access to independent studies“ seems to me like a procedural issue, as does “equal treatment“. And how about what you call “scientificity“: why would this be a recognition justice issue?

Results section

My main comment here is that the whole section would need to be thoroughly rewritten in such a manner that it presents and interprets the results in a manner that is meaningful for the reader. Now, the overwhelming majority of the text consists of a kind of a listing of correlations found between different survey questions/answers, and it is very difficult for the reader to interpret what such correlations actually tell us about justice, and about the interrelations between different aspects and dimensions of justice. Also, the sheer number of highly diverse types of generic concepts such as honesty, transparency, comprehensibility, participation, utilitarianism, retribution, and environmentally burdened regions makes it very difficult to make sense of the findings.

As I mentioned above, the treatment of the issue of “scientificity“ or “scientific decisions“ is problematic. Here, I should simply add that I found the survey questions concerning this topic somewhat strange. If you ask people whether scientific arguments are more important than fears or emotions expressed, who would say that fears and emotions are more important?

Yet another remark on the questions: when you asked whether “political consideration is important for the best repository site“, what did you have in mind, in fact? Were “political considerations“ defined in some way or another? For me, siting decision can only be political, hence the question doesn’t really make sense. But probably the meaning was something else in this survey, and in the minds of the respondents.

Discussion section

"For honesty, the survey has shown that the aspects of vital communication as defined by Habermas [40] are empirically verifiable." - What do you mean by this statement? What are "aspects of vital communication"? I suppose that you refer to Habermas' theory of communicative action, but I don't really grasp the meaning of this sentence.

"To the author's knowledge, only a few studies have been carried out about transparency and honesty in participatory processes, so there is a need for further research." - Was your analysis only about participatory processes? I didn't think so, but this sentence seems to suggest this was the case.

"Although this strengthens the impression that a rational and scientific process is of utmost importance..." - See my earlier comments. Which rationality? Whose rationality? What kind of science? Can a siting process be purely scientific?

"...it shows that the notion raised by Krütli et al. [30] that the participatory process is regarded as a corrective for technocratic decision-making cannot be neglected." - How does it show this? And what is actually meant by "it" in this sentence? Why cannot this notion be neglected? "Regarded" by whom?

"This finding reinforces the qualitative observations from the German site search process by Schwarz et al. [78]." - What were these observations?

"This survey has shown that the participants can differentiate between a deep-geological repository and their above-ground living environment." - What is the meaning here? Seems a bit strange and obvious: why wouldn't most sane people be able to differentiate between a repository and the above-ground living environment?

"A contradiction was found here that cannot be resolved from the available survey data." - I couldn't see what the contradiction here was.

"this question would just lie in the realm of retribution, which in turn would cause a source of injustice." - What is the idea here? Wasn't retribution justice one of the justice dimensions? From where does the "source of injustice" come to the picture here?

"the results would suggest that notions of utilitarianism as presented by Mill and Bentham [46] or Lazari-Radek and Singer [47], or retribution [50] would not apply to nuclear waste management, it seems too quick to dismiss those notions, especially as the site selection process is still in an early phase and feelings of actual affectedness are still rather little. As the site selection will mature and spatial concreteness will increase, utilitarianism needs to be revisited." - I could not follow the argument here. What is the message and meaning of these sentences?

"The analysis of justice of recognition proves the most difficult for a closed survey setting with a low entrance barrier, as political and societal expectations while answering cannot be properly addressed." - Again, I failed to understand what is meant here.

"Especially the role of a science-based decision was the most important aspect of a just repository site, but the quality of this kind of decision remains vague." - Same as above, I don't understand the message. What do you mean when you say that the quality of this kind of decision remains vague? Which decision? Vague in which manner? How can quality be vague?

"Questions of path dependencies and uncertainties need to be addressed to assess if science-based decisions are always perceived as neutral and just..." - Again, what do you actually try to say here? Why would science-based decisions be always perceived as neutral and just? Should they and could they? And the everlasting question: what do you mean by "science-based"? Same as "informed by science"? "Dictated by science"? Which science?

"This can be done by conducting a discourse analysis or a scenario-based survey." - Those are perhaps some among the numerous options for conducting further analysis of perceptions concerning the quality of the needed scientific inputs. Not sure whether it is necessary or useful to single out discourse analysis and scenario-based surveys as methods. Why mention these specific methods?

"which makes a strong perception of justice difficult." - Please clarify to make the meaning understandable. What makes this difficult? What is a "strong perception"? Why is it made difficult?

"This is consistent with Schlosberg’s [20] approach to the concept of environmental justice, as well as Honneth's [57] remarks on how to properly recognize." - Recognise what?? What is the essence of Schlosberg's approach to which you refer here?

Conclusions

"This article thematized the following question: Which aspects of justice are of particular importance for the German repository site to be perceived as just?" - What do you mean by thematising? Did you ask the respondents to prioritise between different justice-related questions and statements?

"Distributional aspects such as utilitarianism, retribution, or exemption of environmentally burdened areas are generally not approved" - As mentioned earlier, here you have a slightly odd selection of issues and elements of highly different nature and, in my opinion, at quite different levels of abstraction. To talk about utilitarianism, retribution and "exemption of environmentally burdened areas" as if they were somehow comparable entities/issues strikes me as highly problematic.

Reviewer 3 Report

The authors have thoroughly revised the manuscript and, in my opinion, the document has improved substantially.
Thank you for considering my observations.